# Teaching Diffusion Models to Ground Alpha Matte

**Tianyi Xiang**                                          *tianxiang6-c@my.cityu.edu.hk*
*Department of Computer Science*
*City University of Hong Kong*

**Weiying Zheng**                                        *weiyingzheng@connect.hku.hk*
*School of Computing and Data Science*
*The University of Hong Kong*

**Yutao Jiang**                                            *smallpeach.gm@gmail.com*
*School of Computer Science and Engineering*
*South China University of Technology*

**Tingrui Shen**                                        *202230040157@mail.scut.edu.cn*
*School of Computer Science and Engineering*
*South China University of Technology*

**Hewei Yu**                                                      *hwyu@scut.edu.cn*
*School of Computer Science and Engineering*
*South China University of Technology*

**Yangyang Xu**[†]                                             *xuyangyang@hit.edu.cn*
*School of Intelligence Science and Engineering*
*Harbin Institute of Technology (Shenzhen)*

**Shengfeng He**[†]                                          *shengfenghe@smu.edu.sg*
*School of Computing and Information Systems*
*Singapore Management University*

**Reviewed on OpenReview:** *https://openreview.net/forum?id=2gNy9Yeg8J*

## Abstract

The power of visual language models is showcased in visual understanding tasks, where language-guided models achieve impressive flexibility and precision. In this paper, we extend this capability to the challenging domain of image matting by framing it as a soft grounding problem, enabling a single diffusion model to handle diverse objects, textures, and transparencies, all directed by descriptive text prompts. Our method teaches the diffusion model to ground alpha mattes by guiding it through a process of instance-level localization and transparency estimation. First, we introduce an intermediate objective that trains the model to accurately localize semantic components of the matte based on natural language cues, establishing a robust spatial foundation. Building on this, the model progressively refines its transparency estimation abilities, using the learned semantic structure as a prior to enhance the precision of alpha matte predictions. By treating spatial localization and transparency estimation as distinct learning objectives, our approach allows the model to fully leverage the semantic depth of diffusion models, removing the need for rigid visual priors. Extensive experiments highlight our model's adaptability, precision, and computational efficiency, setting a new benchmark for flexible, text-driven image matting solutions. The code is available at `https://github.com/xty435768/TeachDiffusionMatting`.

---

[†]Corresponding authors.

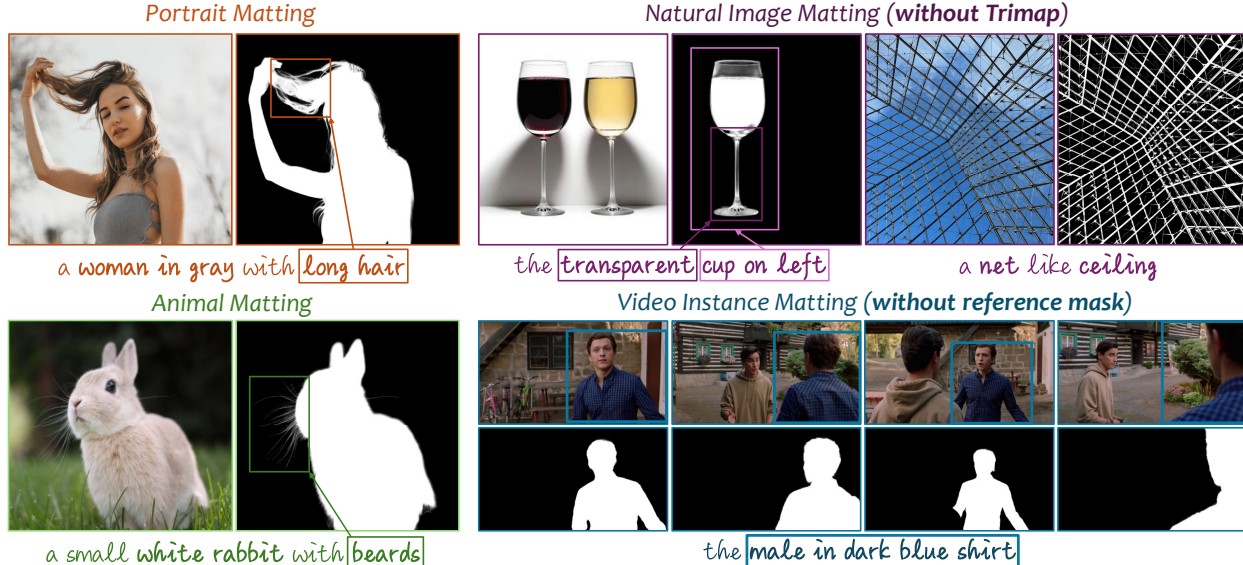

Figure 1: We propose to teach a single diffusion model capable of handling various matting tasks using task-specific text guidance (shown below each sample) with keywords marked in **bold**. By introducing language-driven priors, we unify diverse matting tasks into a soft grounding framework. Our novel pipeline for teaching the text-to-image diffusion model achieves state-of-the-art performance on this challenging problem.

# 1 Introduction

Image matting is a longstanding and foundational task in computer vision, aimed at extracting a foreground object from an image and estimating the transparency of each pixel. Traditionally, this process is modeled by the following equation (Porter & Duff, 1984):

$$I = \alpha F + (1 - \alpha)B, \tag{1}$$

where only the input image $I$ is known, while the alpha matte $\alpha$, foreground $F$, and background $B$ colors are unknowns. Solving this ill-posed problem has led researchers to develop various priors, including trimaps (Levin et al., 2007; Xu et al., 2017; Yao et al., 2024a), background estimates (Lin et al., 2021), binary masks (Yu et al., 2021; Huynh et al., 2024), and user-provided inputs (Ye et al., 2024; Li et al., 2024b).

However, these visual-level priors present limitations in many matting scenarios. For instance, trimaps require substantial annotation efforts, while the background and binary masks are unsuitable for dynamic scenes without temporal information. Additionally, interactive inputs are generally limited to static settings and lack generalizability for complex textures in natural image matting (see the upper-right example in Fig. 1). Thus, there is a strong need for a unified and user-friendly approach.

Recently, visual-language approaches have achieved significant progress in dense visual prediction tasks (Gavrilyuk et al., 2018; Ye et al., 2019; Luo et al., 2020; Wu et al., 2022). Notably, text-to-image diffusion models like Stable Diffusion (SD) (Rombach et al., 2022) have enabled tasks such as open-vocabulary panoptic segmentation (Xu et al., 2023a) and referring image segmentation (Zhao et al., 2023a) by leveraging capabilities for semantic differentiation and cross-modal attention. This is a significant advancement over previous visual-language models like CLIP (Radford et al., 2021), which primarily focus on image-level similarity and do not effectively capture the pixel-level correspondences required for tasks like matting. The rich and powerful internal features along with the interpretable cross-attention maps naturally encode pixel-level visual–language priors, making SD a promising candidate for matte grounding tasks through

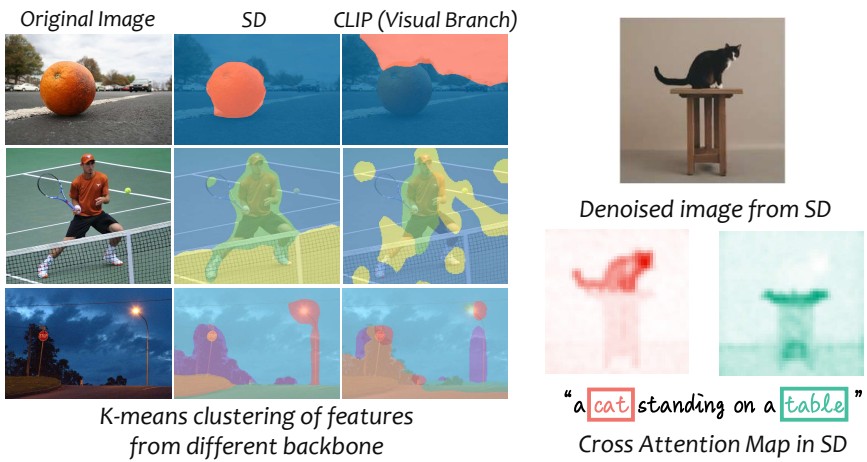

Figure 2: Visualization of features from SD (Rombach et al., 2022) and CLIP (Radford et al., 2021) (left) and cross-attention maps in SD (right). SD exhibits strong semantic differentiation in its features and cross-modal attention maps, with both inter-correlation and intra-consistency.

natural language guidance, combining the expressive and intuitive nature of language with the strong prior knowledge embedded in the SD model. As visualized in Fig. 2, SD's semantic representations show strong differentiation and consistency both within and across object regions, making it especially well suited for the fine-grained demands of advanced alpha matte estimation, which often has to resolve multiple similar objects and complex transparency patterns without relying on rigid priors such as trimaps, masks, or user interactions.

This raises a natural question: *can this prior knowledge be applied effectively to alpha matting?* Unlike standard visual grounding tasks that typically produce binary masks for object identification, alpha matting requires both precise object localization and fine-grained transparency estimation, with the alpha value $\alpha \in [0, 1]$ in Eq. (1) accurately predicted. We define this unified approach as **soft grounding**, addressing both instance-level localization and transparency simultaneously for more adaptable and nuanced matting applications.

While SD holds significant potential, directly adapting it for soft grounding tasks introduces challenges due to the simultaneous requirements of localization and transparency estimation. Semantic localization relies on high-level semantic understanding, whereas transparency estimation depends on fine-grained, low-level details. Jointly optimizing these sub-tasks within a single model often results in performance trade-offs due to their competing objectives. Previous adaptations of SD for portrait matting (Xu et al., 2024a; Wang et al., 2024) rely on the assumption of a prominent, easily distinguishable foreground object, which simplifies localization. However, this assumption is often invalid in more complex scenarios involving multiple objects or intricate transparency patterns, such as those encountered in natural image matting (Fig. 1), which complicate the separation between foreground and background. Therefore, directly applying SD may yield suboptimal results due to the increased difficulty in resolving both localization and transparency.

In this work, we propose a framework to teach SD to ground alpha mattes with any user prompts. Rather than attempting to estimate the alpha matte directly, our approach introduces an intermediate objective that initially guides the model to localize semantic components of the alpha channel. Once this "teacher" model is trained for localization, we introduce a distillation process where a "student" model leverages the teacher's learned semantic information to progressively refine transparency estimation. This sequential framework ultimately enables accurate alpha matte prediction with minimal post-processing.

This approach offers two primary advantages. First, by avoiding continuous fine-tuning of the teacher model for transparency estimation, we preserve much of its pretrained semantic knowledge, maintaining a clear distinction between localization and transparency tasks. Second, the strong semantic foundation established

by the teacher model enables us to employ a more computationally efficient student model, enhancing the practicality of this approach for real-world applications.

Our main contributions are as follows:

- We unify various matting tasks into a soft grounding problem, leveraging the Stable Diffusion model's visual-language capabilities.

- We introduce a distillation framework that teaches SD to ground alpha mattes by disentangling instance-level localization and transparency estimation, enhancing performance in scenarios involving multiple objects or complex transparency.

- This framework enables optimization of the student model's structure, improving computational efficiency for practical applications.

- Extensive evaluations demonstrate that our method outperforms multiple baselines in soft grounding tasks, achieving competitive speed and generalizing well across different matting categories.

## 2 Related Work

### 2.1 Alpha Matte Grounding

Due to the inherent ambiguity in image matting, most existing methods rely heavily on trimaps as prior guidance (Levin et al., 2007; He et al., 2011; Chen et al., 2013; Xu et al., 2017; Hou & Liu, 2019; Park et al., 2022; Wang et al., 2023b; Yao et al., 2024a; Xu et al., 2023b; Hu et al., 2024). However, generating accurate trimaps that can distinguish foreground, background, and unknown regions is costly and time-intensive. Consequently, recent research has focused on exploring alternative priors, including background information (Lin et al., 2021; Sengupta et al., 2020), binary masks (Park et al., 2023; Yu et al., 2021; Huynh et al., 2024; Sun et al., 2022; Li et al., 2024a; Yang et al., 2025), in-context priors (Guo et al., 2024), and interactive inputs such as points, bounding boxes, and scribbles (Wei et al., 2021; Yang et al., 2022; Yao et al., 2024b; Ye et al., 2024; Li et al., 2024b; Xia et al., 2024).

Text-based priors, which offer more flexible and intuitive guidance by dynamically and autonomously identifying matting objects, have also been investigated. For instance, Li *et al.* (Li et al., 2023a) introduce CLIPMat, which is the first text-based matting method leveraging a pre-trained CLIP model (Radford et al., 2021) to integrate visual and textual features for matting. Xu *et al.* (Xu et al., 2024b) further enhance the feature fusion approach based on Li et al. (2023a). However, since CLIP was designed for cross-modal similarity across entire images, it struggles with the high semantic precision and detailed requirements of pixel-level matting. In contrast, our method is based on a text-to-image diffusion model that provides pixel-level visual-language priors and robust semantic differentiation, making it more suited to the nuanced demands of image matting.

### 2.2 Diffusion Models for Image Matting

Diffusion models have shown great potential in a variety of applications, including generative tasks (Rahman et al., 2023; Shi et al., 2024; Blattmann et al., 2023; Brooks et al., 2023; Ruiz et al., 2023; Wang et al., 2023a) and dense prediction tasks (Ji et al., 2023; Zhao et al., 2023a; Xu et al., 2023a; Lee et al., 2024; Xu et al., 2024a; Burgert et al., 2023). Some studies have applied the prior knowledge encoded in these models to image matting tasks.

Guo *et al.* (Guo et al., 2024) leverage the in-context correspondence priors within Stable Diffusion (Rombach et al., 2022), using guidance from a reference image to perform matting of the same object across different scenes. Wang *et al.* (Wang et al., 2024) approach matting as a generative task, applying denoising over multiple time steps and fine-tuning the model to predict the alpha matte. However, these methods depend on forecasting alpha mattes directly, limiting their effectiveness to matting a single object and often resulting in inefficiencies due to the high computational cost of diffusion models.

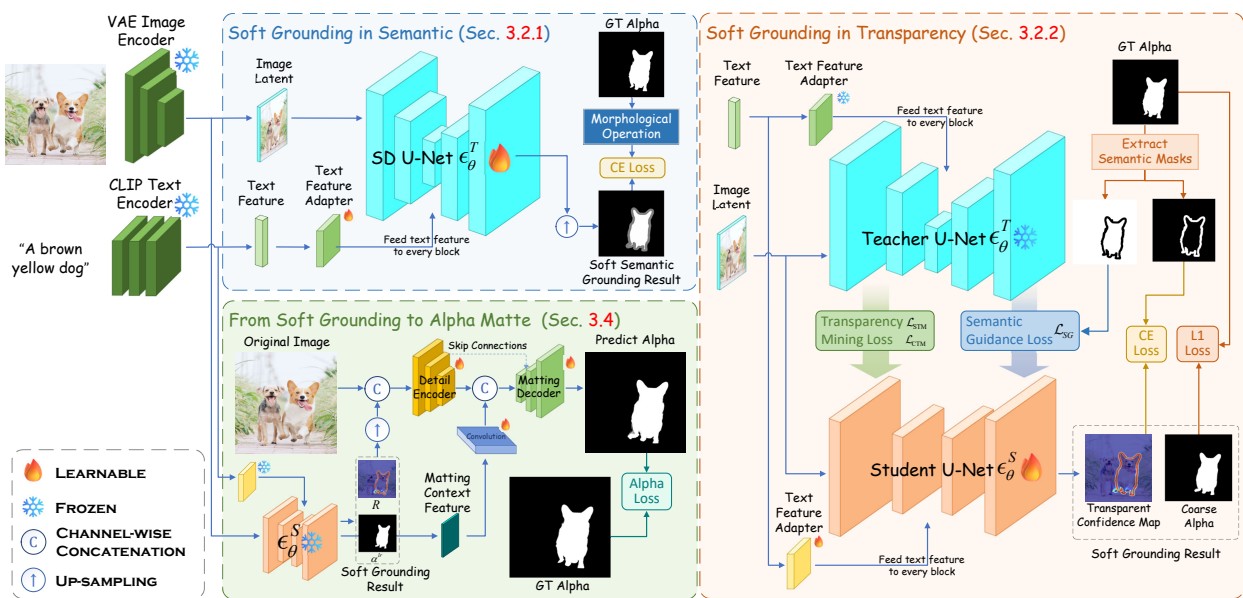

Figure 3: Overview of the pipeline of our model, with unimportant operations and blocks omitted for clearer illustration. Motivated by the goal of disentangling two learning objectives, we first adapt the diffusion model to a sub-objective of instance-wise localization of semantic components of the alpha matte (blue part on top-left). Next, we propose a distillation framework (orange part on the right) to distill the localization information from the teacher model ($\epsilon_\theta^T$) into the student model ($\epsilon_\theta^S$) using the Semantic Guidance Loss ($\mathcal{L}_{SG}$) based on internel features, while encourage $\epsilon_\theta^S$ to simultaneously explore the transparent-related information via the Transparency Mining Loss ($\mathcal{L}_{STM}$, $\mathcal{L}_{CTM}$) based on attention maps. $\epsilon_\theta^S$ learns to predict the coarse alpha matte $\alpha^{lr}$ and the transparent confidence map $R$ to further enhance $\alpha^{lr}$ into $\alpha$ at full resolution (green part on bottom-left).

Alternatively, some methods operate directly in pixel space by diffusing a disturbed trimap (Xu et al., 2023b) or pure noise (Hu et al., 2024) until a clean alpha matte is produced. Li *et al.* (Li et al., 2024c) extend this approach to latent space, introducing modified self-attention to better model matting context. However, these approaches rely heavily on a trimap as conditioning input, limiting their flexibility in practical applications.

# 3 Method

## 3.1 Overview

**Problem Formulation.** Given an input image $I \in \mathbb{R}^{H \times W \times 3}$ and a foreground text expression $\mathcal{T}$, our goal is to teach the Stable Diffusion (SD) model soft grounding and predict the alpha matte $\alpha \in [0,1]^{H \times W \times 1}$ in a single step.

**Network Design.** Our framework is illustrated in Fig. 3. To teach the SD model to ground the alpha matte, we disentangle the soft grounding problem into two sub-objectives using an intermediate objective, soft semantic grounding, along with a distillation framework. First, the original diffusion model is trained to localize the semantic components of the target alpha matte (see Sec. 3.2.1). Then, an asymmetric distillation framework with two tailored objectives guides the model to refine transparency estimation while preserving the localization capability learned in the initial stage (see Sec. 3.2.2). This approach ensures effective task disentanglement. Moreover, the distillation framework enables the adoption of a computationally efficient model for soft grounding, improving its practicality (see Sec. 3.3). Using the soft grounding results and output features from the well-taught diffusion model, the matting decoder can readily predict the final full-resolution alpha matte (see Sec. 3.4). We discuss our proposed paradigm below.

### 3.2 Soft Grounding Teaching

#### 3.2.1 Soft Grounding in Semantic

We first teach the diffusion model to address an intermediate task: localizing each semantic component of the target alpha matte. This is achieved by fine-tuning the denoising U-Net ($\epsilon_\theta$) starting from the original weights of SD. Specifically, taking the latent code $z_I := \mathcal{E}(I)$ at a resolution of $\frac{H}{8} \times \frac{W}{8}$ and the corresponding linguistic features $f_\mathcal{T}$ as conditional input, $\epsilon_\theta$ is fine-tuned to distinguish the semantic regions of alpha matte (the foreground, background, and transparent regions) and to predict classification results for each region. The results are represented as $s_I \in \{0,1\}^{\frac{H}{8} \times \frac{W}{8} \times 3}$ of $I$. To achieve this, we replace the last latent prediction head in the original $\epsilon_\theta$ with a new head comprising two groups of ConvBNReLU layers to predict $s_I$. The linguistic features $f_\mathcal{T}$ are generated by the frozen CLIP text encoder (Radford et al., 2021) and further refined using a two-layer MLP text adapter, inspired by Gao et al. (2024). We train the model to predict classification probabilities $p_I \in [0,1]^{\frac{H}{8} \times \frac{W}{8} \times 3}$ for each channel in $s_I$ by minimizing the cross-entropy loss $\mathcal{L}_{CE}$:

$$\mathcal{L}_{\text{CE}} = -\sum_{c=1}^{3} s_{GT}^c \log(p_I^c), \tag{2}$$

where $s_{GT}$ represents the GT region representation, obtained by applying morphological operations to the GT alpha, and $p_I^c$ denotes the classification probability for the $c$-th channel in $s_I$. We denote the fine-tuned $\epsilon_\theta$ as $\epsilon_\theta^T$.

#### 3.2.2 Soft Grounding in Transparency

After the convergence of training $\epsilon_\theta^T$, our next goal is to further explore transparency information based on $\epsilon_\theta^T$ until achieving soft grounding. A straightforward approach would be to fine-tune $\epsilon_\theta^T$ to predict transparency directly. However, this presents two key challenges. First, fine-tuning $\epsilon_\theta^T$ for transparency estimation risks disrupting its well-learned semantic knowledge. Second, the computational cost of $\epsilon_\theta^T$ is typically high, limiting its practicality for soft grounding. To address both challenges, we propose a novel soft grounding distillation framework. This framework fully exploits the strong semantic guidance embedded in the intermediate features of the pre-trained $\epsilon_\theta^T$ while enabling the student model ($\epsilon_\theta^S$) to refine transparency estimation in the corresponding alpha matte. As a result, $\epsilon_\theta^S$ is expected to gain full capabilities for solving the soft grounding problem. For implementation, $\epsilon_\theta^S$ is trained to predict a coarse alpha matte $\alpha^{lr} \in [0,1]^{\frac{H}{8} \times \frac{W}{8} \times 1}$ and a transparent confidence map $R \in \mathbb{R}^{\frac{H}{8} \times \frac{W}{8} \times 1}$, which indicates regions that may be transparent. These outputs are then used for further upsampling and refining $\alpha^{lr}$ to $\alpha$. Next, we introduce two key losses to effectively guide this distillation process.

**Strong Semantic Guidance Loss.** Inspired by Kim et al. (2024), we introduce a feature-level distillation loss $\mathcal{L}_{SG}$, to ensure that the semantic representation of the student model $\epsilon_\theta^S$ closely aligns with that of the teacher model $\epsilon_\theta^T$, at an intermediate feature level, formulated as:

$$\mathcal{L}_{\text{SG}} = \sum_l \|M \odot F_T^l - M \odot \Phi(F_S^l)\|_2^2, \tag{3}$$

where $F_T^l$ denotes the intermediate feature of $\epsilon_\theta^T$ at layer $l$ (and similarly, $F_S^l$ for the student model). Here, $M$ is a binary mask with foreground and background regions set to 1 and other regions set to 0. $M$ ensures that semantic supervision dose not interfere with the exploration of transparency details. The symbol $\odot$ represents the Hadamard product. The projection module $\Phi$ aligns the intermediate features of the two models, accounting for differences in their feature spaces due to distinct learning objectives. It consists of 3 learnable convolutional layers (Cv) and an intermediate LeakyReLU layer, defined as:

$$\Phi(F) = \text{Cv}_{3\times3}(\text{Cv}_{3\times3}(\text{LeakyReLU}(\text{Cv}_{1\times1}(F)))). \tag{4}$$

**Transparency Mining Loss.** To further encourage $\epsilon_\theta^S$ to explore transparency details, we aim to extract and mine potential transparency information from $\epsilon_\theta^T$. We begin with the self-attention maps of $\epsilon_\theta^T$, a critical component for modeling intra-object consistency. Since $\epsilon_\theta^T$ possesses well-learned semantic knowledge of alpha, the implicit transparency information used to identify such regions can serve as transparency cues,

guiding $\epsilon_\theta^S$ to further refine transparency details. Given the self-attention map $\mathbf{A}_f := \mathrm{SoftMax}(\frac{\mathbf{Q}_f \cdot \mathbf{K}_f}{\sqrt{d}})$ derived from feature $f$, we introduce a self-attention-based transparency mining loss $\mathcal{L}_{\mathrm{STM}}$, to constrain affinities specifically related to the alpha matte. To accomplish this, we propagate the GT alpha, $\alpha_{GT}$, using the averaged self-attention map, $\mathbf{A}_{SA}$, computed across all self-attention layers at the same resolution. This propagation is performed via matrix multiplication ($\mathbf{A}_{SA} \otimes \alpha_{GT}$). We then minimize the difference between the propagated results of the teacher and student models. The rationale is as follows: $\mathbf{A}_{SA}$ acts as a transition matrix, while $\alpha_{GT}$ represents the current state. Their matrix multiplication transforms the current state into a resulting state. By constraining this resulting state, we indirectly regulate the $\alpha$-related affinities within the self-attention map. This approach enhances the exploration of transparency details without incurring significant computational overhead, as it avoids constraining the entire self-attention map. We formulate $\mathcal{L}_{\mathrm{STM}}$ as follows:

$$\mathcal{L}_{\mathrm{STM}} = ||\mathbf{A}_{SA}^T \otimes \alpha_{GT} - \mathbf{A}_{SA}^S \otimes \alpha_{GT}||_2^2, \tag{5}$$

where $\otimes$ denotes matrix multiplication (not element-wise multiplication). Note that $\alpha_{GT}$ is down-sampled to the corresponding resolution using bilinear interpolation to satisfy the requirements for $\otimes$. We also extract potential transparency cues from the text expression using a similar cross-attention map distillation loss $\mathcal{L}_{\mathrm{CTM}}$, which directly constrains the cross-attention maps between $\epsilon_\theta^S$ and $\epsilon_\theta^T$, formulated as:

$$\mathcal{L}_{\mathrm{CTM}} = ||\mathbf{A}_{CA}^T - \mathbf{A}_{CA}^S||_2^2. \tag{6}$$

For implementation, all attention maps are grouped by resolution and averaged to compute the loss. The final attention distillation loss is then calculated as the mean of the losses across all resolutions.

**Total Objectives.** In addition to the two key losses described earlier, we incorporate an L1 loss to train $\alpha^{lr}$ and a binary cross-entropy (BCE) loss to train $R$. These are defined as follows:

$$\mathcal{L}_{\alpha^{lr}} = ||\alpha^{lr} - \alpha_{GT}^{lr}||_1, \tag{7}$$

$$\mathcal{L}_R = -\widehat{M} \log(\frac{1}{1+e^R}) - (1 - \widehat{M}) \log(\frac{e^R}{1+e^R}), \tag{8}$$

where $\alpha_{GT}^{lr}$ represents the down-sampled GT alpha matte, and $\widehat{M} := 1 - M$ is a binary mask that indicates transparent regions. The final objective, $\mathcal{L}_{\epsilon_\theta^S}$, used to train $\epsilon_\theta^S$, is formulated as:

$$\mathcal{L}_{\epsilon_\theta^S} = \lambda_{\mathrm{STM}}\mathcal{L}_{\mathrm{STM}} + \lambda_{\mathrm{CTM}}\mathcal{L}_{\mathrm{CTM}} + \lambda_{\mathrm{SG}}\mathcal{L}_{\mathrm{SG}} + \lambda_{\alpha^{lr}}\mathcal{L}_{\alpha^{lr}} + \lambda_R\mathcal{L}_R, \tag{9}$$

where the $\lambda$ coefficients are hyperparameters that balance the contribution of each loss term.

### 3.3 Structural Optimization on $\epsilon_\theta^S$

In our distillation framework, $\mathcal{L}_{\mathrm{SG}}$ provides strong supervision signals to preserve localization knowledge by directly minimizing discrepancies in the semantic content of internal features. This highlights a key advantage of our approach: achieving a balance between maintaining performance and enhancing computational efficiency (see Sec. 3.2.2). To fully exploit this advantage, we apply two primary structural optimization techniques to the vanilla $\epsilon_\theta^S$ (which initially shares the same architecture and parameters as $\epsilon_\theta^T$): block pruning and self-attention optimization.

**Block Pruning.** We prune redundant blocks from the vanilla student model following Kim et al. (2024), resulting in a model with fewer parameters. Guided by $\mathcal{L}_{\mathrm{SG}}$, the performance degradation caused by this pruning is effectively mitigated. Additionally, the other loss terms encourage the student model to further assimilate soft grounding knowledge.

**Self-Attention Optimization.** The self-attention mechanism in Transformer blocks is a critical component for modeling long-range dependencies. However, the matrix multiplication operations in self-attention are computationally expensive, resulting in an overall time complexity of $O(n^2 d + n d^2)$, which becomes particularly burdensome at higher resolutions. Nevertheless, the soft grounding task has more relaxed accuracy requirements, as the learned outputs ($\alpha^{lr}$ and $R$) provide sufficient flexibility to tolerate errors in the subsequent alpha enhancement process. Inspired by this, we propose that the self-attention operation can be optimized into a more computationally efficient form by learning asymmetric sparse correspondences.

Specifically, we argue that the dense affinity matrix derived from the self-attention map $\mathbf{A}_f$ can be simplified into a unidirectional sparse representation. To achieve this, we can learn a smaller set of representative feature tokens and use their affinities to approximate the affinities between the original feature tokens and others within a relatively small region. Thus, we introduce a learnable down-sampling operation $\phi$, which is applied to $f$ when calculating $\mathbf{K}$ and $\mathbf{V}$ (i.e., $\mathbf{K}^S = W_K\phi(f), \mathbf{V}^S = W_V\phi(f)$), while $\mathbf{Q}$ remains unchanged. $\phi$ can be implemented using a single convolution layer with a kernel size and stride of $k \times k$, where $k$ is chosen from $\{2^i | 1 \leq i \leq \log_2 h_{\mathbf{A}_f^S}\}$, and $h_{\mathbf{A}_f^S}$ denotes the height of the square matrix $\mathbf{A}_f^S$, theoretically. The quantitative analysis of the reduction in computational cost can be found in the **Appendix**. Note that $\mathcal{L}_{\text{STM}}$ contributes to preserving intra-object cohesion and textural information within the optimized self-attention mechanism.

### 3.4 From Soft Grounding to Alpha Matte

To up-sample $\alpha^{lr}$ from a resolution of $\frac{H}{8} \times \frac{W}{8}$ to $\alpha$ at $H \times W$, we adopt the detail encoder-decoder structure proposed in Yao et al. (2024a). The encoder takes the concatenation of $(I, \alpha^{lr}, R)$ as input to extract detail features. Before that, $\alpha^{lr}$ and $R$ are interpolated to match the resolution of $I$, and $R$ is processed through a sigmoid function. The detail features from the encoder's final layer are concatenated with features extracted from $\epsilon_\theta^S$ and then passed to a decoder to predict the final alpha matte $\alpha$. For more efficient training, we freeze $\epsilon_\theta^S$ and train only the parameters of the encoder and decoder. Additionally, a single learnable ConvBNReLU block is applied to the output features from $\epsilon_\theta^S$ before concatenation to align the feature spaces. This process is supervised using an L1 loss and a Laplacian loss following Li et al. (2022).

**Extend to High-resolution Inference.** We implement a simple upsampling module based on sparse convolution (Contributors, 2022), integrated with the matting decoder, to enable inference at arbitrary high resolutions (up to 2K). This sparse convolution approach effectively reduces computational cost and memory usage during the upsampling process.

## 4 Experiment

### 4.1 Implementation Details

**Data Acquisition.** The data used to train our model comprises 4 matting datasets (RefMatte (Li et al., 2023a), P3M10K (Li et al., 2021a), AM2K (Li et al., 2022), RM1K (Wang et al., 2023b)), and 1 grounding segmentation dataset (RefCOCO (Kazemzadeh et al., 2014)). Considering there are no text annotations for P3M10K, AM2K, and RM1K, we adopt BLIP2 (Li et al., 2023b) to generate text annotations for each sample in these datasets by guiding the BLIP2 model to describe the appearance of the object in the image. For samples from RefMatte and RefCOCO during training, we randomly select one expression for each object if multiple expression annotations exist for the same object. Since there are no matting-level annotations for RefCOCO, we first generate the pseudo trimap according to the mask annotations using morphological operations, then use Yao et al. (2024a) to obtain alpha annotations.

**Training Details.** All stages of our model's training process adopt a consistent data scheduling strategy. Specifically, we train the model on RefMatte during odd-numbered iterations and on RefCOCO in every even iteration. We also insert a special iteration after every 4 iterations to perform training on P3M10K, AM2K, and RM1K. We randomly select 1 dataset among the 3 to train our model in this special iteration. We set the kernel size of the morphological operation to 15, and we set $(\lambda_{\text{STM}}, \lambda_{\text{CTM}}, \lambda_{\text{SG}}, \lambda_{\alpha^{lr}}, \lambda_R)$ to $(10, 0.1, 0.5, 10, 1)$. The timestep input of the SD model is set to 1.0 during both training and inference, which is consistent with previous works (Zhao et al., 2023a; Lee et al., 2024; Xu et al., 2024a). This setting can transform the multi-step SD model into a deterministic single-step perception model effectively. Other training settings, including batch size, learning rate, total iterations, and rationales behind setting $\lambda$s, can be found in the **Appendix**.

**Details about Optimized $\epsilon_\theta^S$.** To thoroughly evaluate our proposed framework, we adopt the most extreme pruning scheme proposed in Kim et al. (2024) to build $\epsilon_\theta^S$, i.e. the *tiny* setting, which removes 5 blocks in the encoder and 5 blocks in the decoder, and removes the entire middle block in $\epsilon_\theta$. For optimizing self-attention

(SA), we directly remove all SA layer at resolution of $64 \times 64$ following Zhao et al. (2023b), and set the $k$ value for SA layers at $32 \times 32$ and $16 \times 16$ ($k_{32^2}$, $k_{16^2}$) as $k_{32^2} = k_{16^2} = 2$.

## 4.2 Evaluate Metrics

Following Rhemann et al. (2009), we adopt 4 matting metrics for evaluation, including SAD, MSE, Gradient Error (GRAD), and Connectivity Error (CONN). These metrics are scaled by $10^3$, $10^{-3}$, $10^3$, $10^3$, respectively. Lower values indicate better performance across all metrics.

## 4.3 Comparison on Soft Grounding

**Baselines.** To comprehensively evaluate our model's performance, we compare it with three types of baselines.

- **Text-guided matting methods.** We select CLIPMat (Li et al., 2023a) as a fundamental baseline, which solves the soft grounding problem directly and shares similar settings with ours. Since CLIPMat is currently closed-source, we re-implement it and use the same training settings as ours.

- **Visual grounding with interactive matting methods.** We select the representative GroundingDINO (GDINO) (Liu et al., 2024b) as the visual grounding method, which produces the bounding box of the foreground object according to the text input. Then we select 3 recent interactive matting methods to obtain the alpha given the bounding box, including MatAny (Yao et al., 2024b), MAM (Li et al., 2024b), and SmartMat (Ye et al., 2024).

- **Grounding segmentation with mask-guided matting methods.** We also select the latest grounding segmentation method PSALM (Zhang et al., 2024) and the SD-based referring segmentation method RefVPD (Zhao et al., 2023a) to generate a mask for the foreground object based on text input. We then apply two mask-guided matting methods to derive the alpha matte from the mask, including MaGGIe (Huynh et al., 2024) and MGMat (Yu et al., 2021).

For fairness, all baselines with cascaded structure, including the visual grounding part (except PSALM and GDINO) and matting part, are aligned to our training data through fine-tuning. Note that PSALM and GDINO are trained on very large-scale datasets. We attempted to fine-tune them directly on our relatively small-scale training dataset, but this yielded poorer results. Consequently, we use their original weights for further comparison.

Note that among all matting components in the baselines (including MatAny, MAM, SmartMat, MGMatting, and MaGGIe), only MaGGIe was trained for human matting exclusively, while the others natively support natural image matting. Although MaGGIe is trained for human matting, its network architecture does not incorporate specific optimizations tailored exclusively for human subjects. Furthermore, the authors of MaGGIe demonstrated its strong zero-shot generalization capabilities on non-human matting tasks in their paper, indicating that MaGGIe can adapt to natural image matting to some extent.

**Benchmarks.** We apply two referring natural matting benchmarks, including RefMatte-Test (Li et al., 2023a) and RefMatte-RW100 (Li et al., 2023a), for soft grounding evaluation. Here, the former is a composition dataset (6,243 instances among 2,500 images) and the latter is a real-world dataset (221 instances among 100 images). Every instance in these two benchmarks has 4 different expressions, so we evaluate all baselines and ours using all expressions and report the average result among these 4 expressions. During evaluation, the input resolution for all methods is set to $512 \times 512$, and the metrics are also calculated on this resolution. We also report the average inference time per sample in milliseconds, using the same machine with a single RTX 3090.

**Quantitative Results.** We show the quantitative comparison on soft grounding in Tab. 1. We found a significant performance gap between CLIPMat and ours since the feature and cross-modal prior within SD model make it easier to solve the soft grounding problem compared with CLIP. Our model also achieves the best performance on most of the metrics compared with the cascaded baselines, even when training data is aligned

Table 1: **Comparison on soft grounding task.** Our model outperforms all baselines on two referring natural image matting benchmarks (RefMatte-Testset and RefMatte-RW100) with faster inference time, demonstrating the effectiveness and practicality of our method.

| Methods | | RefMatte-Testset | | | | RefMatte-RW100 | | | | Inference |
| | | SAD↓ | MSE↓ | GRAD↓ | CONN↓ | SAD↓ | MSE↓ | GRAD↓ | CONN↓ | Time↓ (ms) |
| --- | --- | --- | --- | --- | --- | --- | --- | --- | --- | --- |
| Grounding Model + Interactive Matting | GDINO+MatAny | 15.52 | 0.0561 | 8.41 | 3.61 | 19.92 | 0.0727 | 9.57 | 7.71 | 984.57 |
| | GDINO+MAM | 15.16 | 0.0552 | 8.92 | 4.59 | 16.96 | 0.0626 | 8.56 | 10.78 | 496.30 |
| | GDINO+SmartMat | 11.69 | 0.0403 | 7.91 | 1.78 | 16.91 | 0.0616 | 9.30 | 5.54 | 122.29 |
| Gounding Segmentation Model + Mask-guided Matting | PSALM+MaGGIe | 8.71 | 0.0299 | 7.63 | 2.67 | 9.90 | 0.0349 | 8.24 | **3.75** | 287.79 |
| | PSALM+MGMat | 8.55 | 0.0301 | 7.02 | 2.57 | 9.41 | 0.034 | 6.83 | 3.81 | 272.21 |
| | RefVPD+MaGGIe | 9.01 | 0.0308 | 8.36 | 3.23 | 11.41 | 0.0399 | 9.64 | 5.63 | 220.79 |
| | RefVPD+MGMat | 8.98 | 0.0315 | 7.89 | 3.33 | 10.32 | 0.0374 | 7.66 | 5.37 | 205.21 |
| Soft Grounding | CLIPMat | 26.56 | 0.1181 | 14.84 | 8.15 | 35.12 | 0.1951 | 21.01 | 18.97 | 102.74 |
| | Ours | **3.19** | **0.0098** | **3.55** | **1.69** | **7.37** | **0.0264** | **6.55** | 5.31 | **95.14** |

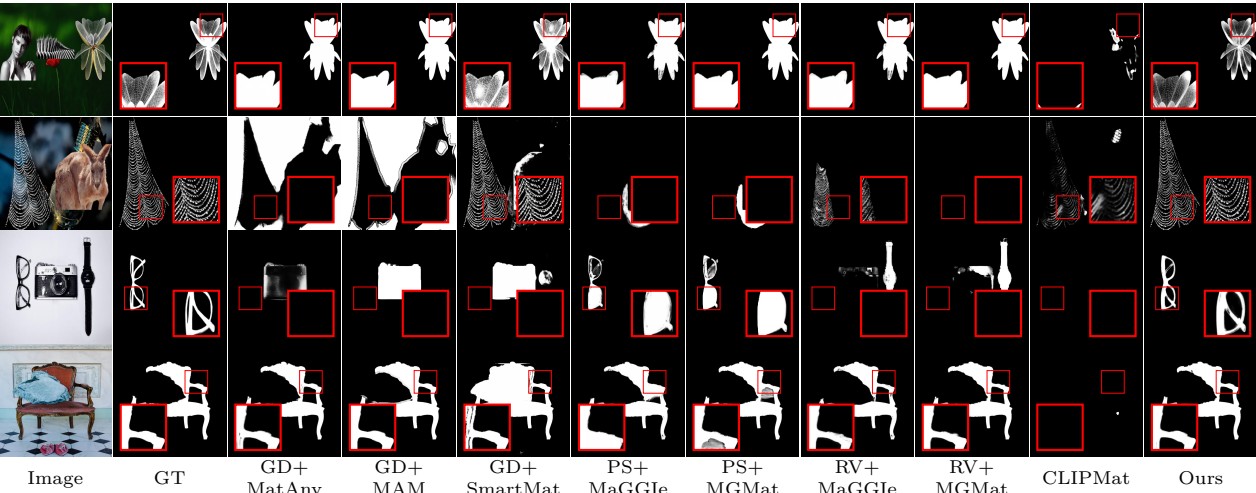

Figure 4: **Qualitative comparison on soft grounding task.** The text inputs from top to bottom are: 1) *the insect which is darkgray*; 2) *the white and non-salient and transparent net*; 3) *a modern-designed glass with a black frame*; 4) *the red chair made of wood*.

among all matting networks. Notably, with all training data aligned, our model still achieves significant performance improvement comparing with two strong baselines (RefVPD+MaGGIe and RefVPD+MGMat). This evidence highlights that the performance gains primarily stem from our novel soft grounding problem modeling and the distillation framework, demonstrating the importance of our proposed algorithm to the performance gain. Moreover, although adopting PSALM, which is based on Large Multi-modal Model (LMM) for grounding segmentation, can obtain lower connectivity error on RW100 benchmark, the other three metrics are still worse than ours and have relatively high inference time. Instead, our framework can teach a structure-optimized model to achieve the best performance with lower inference time, showing the effectiveness of our proposed framework.

**Qualitative Results.** Some qualitative comparisons are shown in Fig. 13. Without properly modeling soft grounding, cascading-based baselines often perform poorly with incorrect semantic and low-quality details as shown in the first two rows. Although some baselines adopt SAM (Kirillov et al., 2023) or LMM (Liu et al., 2024a), they still show sub-optimal performance on soft grounding in the real-world (last two rows). More comparison results can be found in the **Appendix**.

## 4.4 Comparison on Generalization Ability

Introducing text prior can unify various matting tasks into soft grounding. To evaluate the generalization ability of our model on such fine-grained matting tasks, we select several specialist methods in their own task to compare with our model quantitatively (Tab. 2). All baselines here are directly applied using the officially released weight, and our model is tested without task-specific tuning. All the metrics are calculated in full

Table 2: **Quantitative comparison on generalization ability across different matting tasks.** Top-2 results are marked in **bold** and underlined. Task-specific methods only perform well on their own task, while ours can generalize to various matting tasks using text-based guidance **without fine-tuning**.

| Method | Type | AM2K (animal) | | P3M-P (portrait) | | RefMatte-Test (referring natural) | |
|---|---|---|---|---|---|---|---|
| | | SAD↓ | MSE↓ | SAD↓ | MSE↓ | SAD↓ | MSE↓ |
| GFM (Li et al., 2022) | animal | **12.08** | **0.0035** | 347.44 | 0.2001 | 239.33 | 0.1335 |
| P3M-ViTAE (Ma et al., 2023) | portrait | 40.43 | 0.0204 | **6.59** | **0.0015** | 290.05 | 0.1616 |
| GenPercept (Xu et al., 2024a) | portrait | 19.04 | 0.0049 | 11.02 | 0.0025 | 269.50 | 0.1456 |
| AIM (Li et al., 2021b) | natural | 28.25 | 0.0101 | 45.41 | 0.0207 | 336.76 | 0.1840 |
| Ours | | 13.81 | 0.0045 | 9.53 | 0.0030 | **24.19** | **0.0112** |

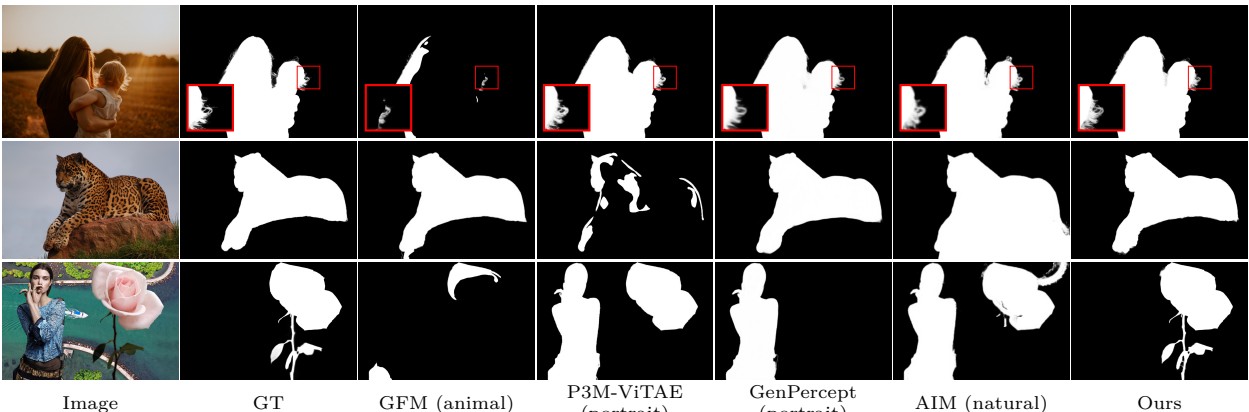

Figure 5: **Qualitative comparison on generalization ability across different matting tasks.** The text inputs used in ours are (from top to bottom): 1) *mother and daughter in field at sunset*; 2) *a jaguar is sitting on top of a rock*; 3) *the works which are thistle and non-transparent.*

resolution. For all test samples in these benchmarks, we fixed the prompt input of ours as "the foreground person" for P3M-P and "the foreground animal" for AM2K. This ensures that the input conditions for our model and the baselines are consistent, enabling a fair comparison. We found that although the baselines excel in their designated tasks, they often lack generalization and struggle with referring matting across diverse categories. In contrast, our model, leveraging a text prior, generalizes effectively to various matting tasks, though it slightly lags behind task-specific experts in their respective domains. Most importantly, our model shows unrivaled performance on referring natural matting tasks, where other baselines falter. We also showcase some qualitative results in Fig. 5. We found GFM and P3M lack generalization, while GenPercept and AIM predict over-smoothed alpha. All of them have poor performance on the soft grounding task.

## 4.5   Ablation Studies

To fully demonstrate the necessity of our key design and modules in our model, we conduct following ablation studies via different training settings and quantify results in Tab. 3, using RefMatte-RW100 as the benchmark.

**Problem Disentanglement.** First, directly training SD for soft grounding proves highly challenging, resulting in suboptimal performance (row 1) compared to a disentangled formulation.

**Knowledge Distillation (KD).** Next, we explore a two-stage approach where SD is pre-trained for localization and then fine-tuned for soft grounding (row 2). This significantly improves performance but remains inferior to our complete KD framework (row 5). These results highlight the importance of preserving semantic knowledge and explicitly separating the two sub-objectives. Furthermore, fine-tuning a lightweight $\epsilon_\theta^S$ without KD leads to a substantial performance drop (row 3), underscoring the critical role of KD in maintaining a balance between efficiency and performance.

Table 3: **Ablation studies.** For instance, ($\epsilon_\theta^T$ 🔥, SG) means training $\epsilon_\theta^T$ on soft grounding task. "$\epsilon_\theta^T$❄️-$\epsilon_\theta^S$ 🔥" denotes the distillation framework. $\epsilon_\theta^S$ is structural optimized to lightweight in row 2~6.

| # | Training settings (SG=Soft Grounding, Local.=Localization) | Φ | Mat. E-D | SAD↓ | MSE↓ | GRAD↓ | CONN↓ |
|---|---|---|---|---|---|---|---|
| 1 | ($\epsilon_\theta^T$ 🔥, SG) | | | 17.50 | 0.0645 | 9.45 | 8.27 |
| 2 | ($\epsilon_\theta^T$ 🔥, Local.), then fine-tune ($\epsilon_\theta^T$ 🔥, SG) | | | 9.75 | 0.0375 | 7.89 | 6.48 |
| 3 | ($\epsilon_\theta^T$ 🔥, Local.), then fine-tune ($\epsilon_\theta^S$ 🔥, SG) | | | 13.27 | 0.0487 | 9.01 | 6.73 |
| 4 | ($\epsilon_\theta^T$ 🔥, Local.), ($\epsilon_\theta^T$❄️-$\epsilon_\theta^S$ 🔥, SG) | | | 15.96 | 0.0534 | 9.26 | 7.55 |
| 5 | ($\epsilon_\theta^T$ 🔥, Local.), ($\epsilon_\theta^T$❄️-$\epsilon_\theta^S$ 🔥, SG) | ✓ | | 7.63 | 0.0267 | 6.98 | 5.35 |
| 6 | ($\epsilon_\theta^T$ 🔥, Local.), ($\epsilon_\theta^T$❄️-$\epsilon_\theta^S$ 🔥, SG) | ✓ | ✓ | 7.37 | 0.0264 | 6.55 | 5.31 |
| 7 | ($\epsilon_\theta^T$ 🔥, Local.), ($\epsilon_\theta^T$❄️-$\epsilon_\theta^S$(w/o. opti.)🔥, SG) | ✓ | ✓ | 6.30 | 0.0223 | 5.52 | 4.33 |

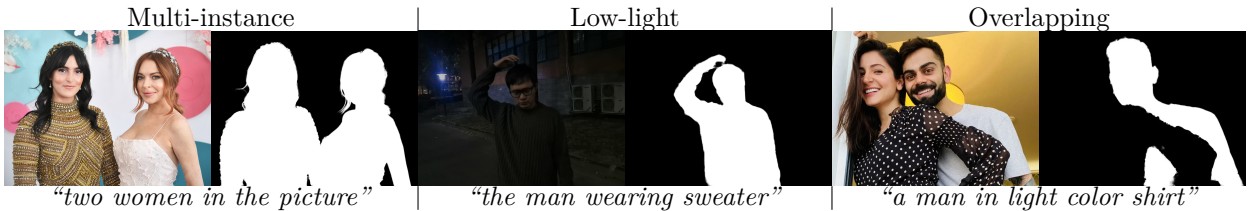

| Multi-instance | Low-light | Overlapping |
|---|---|---|
| *"two women in the picture"* | *"the man wearing sweater"* | *"a man in light color shirt"* |

Figure 6: **Robustness Evaluation.** Our method demonstrates strong performance across a variety of challenging scenarios.

**Other Network Components.** We also observe that removing the feature projection module (Φ) confuses $\epsilon_\theta^S$ during distillation, as $\epsilon_\theta^T$ and $\epsilon_\theta^S$ have distinct learning objectives, leading to degraded performance (row 4). Additionally, integrating a dedicated matting encoder and decoder (Mat. E-D) (row 6, complete framework) further improves the quality of the predicted alpha matte.

**Structural Optimization.** Lastly, we evaluate the impact of structural optimization. While the vanilla $\epsilon_\theta^S$ without structural optimization (row 7) achieves higher performance, its inference time is significantly longer (143ms per image under the same setting in Sec. 4.3). Although there is a slight performance degradation, our framework effectively teaches the lightweight model to achieve impressive performance with a much lower inference time, it is important to note that our optimized model still outperforms most baseline methods, as demonstrated in our evaluations (see Tab. 1), validating the effectiveness of our framework in modeling soft grounding. Furthermore, the optimized student model achieves a significant 33.46% reduction in inference time (from 143ms to 95.14ms). This reduction is particularly valuable in practical applications, such as mobile devices, where lower computational requirements enhance user experience. In future work, we aim to explore more advanced optimization strategies to achieve real-time matting while further minimizing performance degradation.

Moreover, with the same number of parameters, comparing with simply inheriting the SD model (row 1), our complete framework (row 7) achieves a significant performance improvement, demonstrating the importance of our proposed algorithm to the performance gain.

More ablation studies on hyperparameters, objectives, and SD versions are provided in the **Appendix**.

### 4.6 Robustness Evaluation and Limitations

Given accurate and well-structured text prompts, our model demonstrates strong robustness against various typical disturbances (see Fig. 6). However, like all prompt-based systems (Zhang et al., 2025; Wu et al., 2025; Qu et al., 2025; Wen et al., 2025), it is susceptible to vague or ambiguous prompts, which can degrade performance (see Fig. 7). Future work could explore prompt engineering techniques or adaptive guidance strategies to enhance robustness and flexibility across a broader range of matting scenarios.

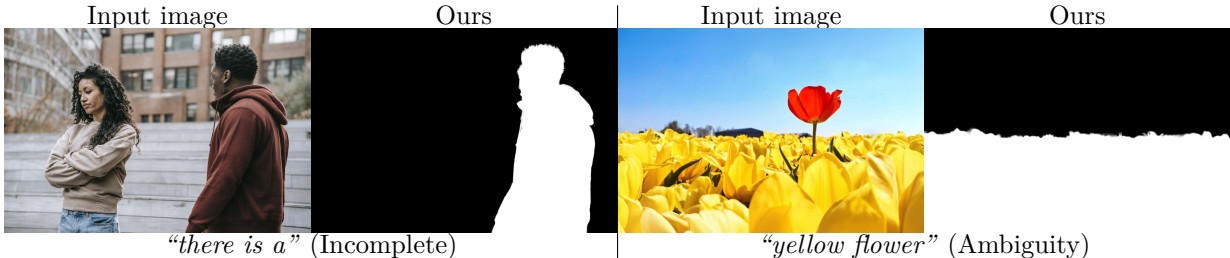

| Input image | Ours | Input image | Ours |

*"there is a"* (Incomplete)        *"yellow flower"* (Ambiguity)

Figure 7: **Some failure cases of our method caused by ambiguous prompts.**

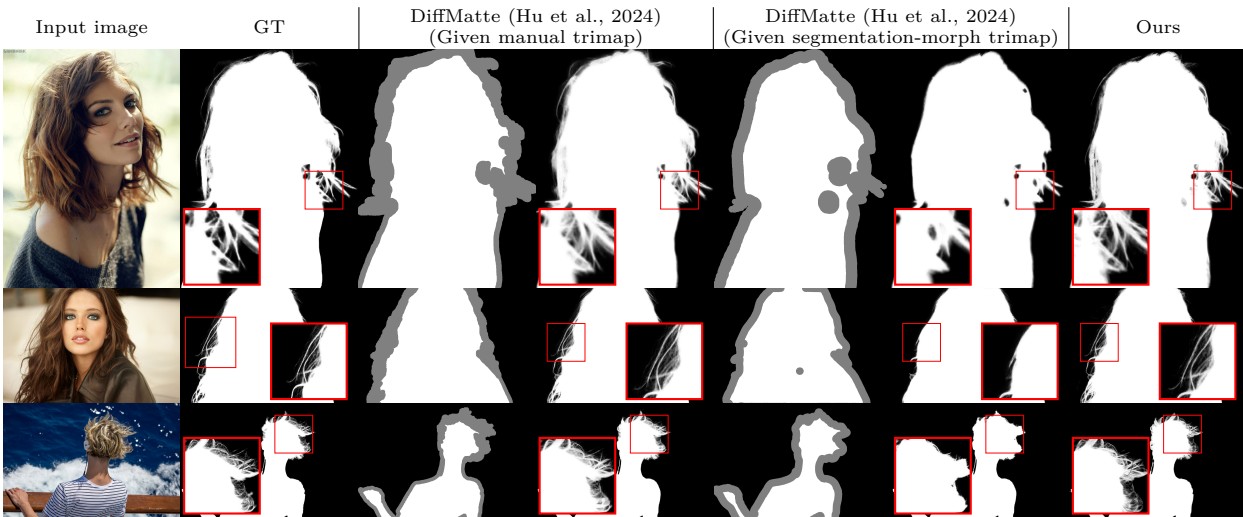

Figure 8: **Qualitative comparison with pixel-space diffusion matting method DiffMatte.** The text inputs of ours are (from top to down): 1) *a beautiful young woman with a short wavy hair*; 2) *a beautiful young woman with long brown hair and blue eyes*; 3) *a woman with headphones on looking out at the ocean*. DiffMatte heavily relies on carefully annotated manual trimaps to produce accurate alpha mattes, while our method only needs text prompts and can achieve comparable performance, despite the latent-space diffusion structure may lead to minor detail loss.

Besides, inheriting biases from pretrained Stable Diffusion models is another inherent limitation of our approach. These biases may impose an upper bound on our model's ability to interpret complex prompt content, as this capability is constrained by the pretrained model's understanding of such prompts. For instance, SD models often struggle with concepts requiring precise counting, such as prompts like "the eleventh person from the left", leading to difficulties in accurately interpreting such positional information. Additionally, biases may adversely affect performance in handling images with challenging conditions, such as those with highly complex lighting or shadows, potentially limiting the general applicability of our model. These limitations stem from the SD model itself. However, we emphasize that the primary focus of our work is not to perfect the diffusion backbone but to effectively adapt it for a novel task. This distinction underscores the unique and significant contributions of our approach in leveraging pretrained models for targeted applications. We also note that future advancements in vision-language models with advanced both prompt and visual understanding capabilities could further enhance progress in soft grounding tasks, and we look forward to such developments. In addition, some structures in SD designed specifically for generation tasks (such as downsampling in VAE) may cause accuracy loss in matting tasks, although our Stage 3 is designed to alleviate this as much as possible. However, compared to pixel-space diffusion matting models (e.g., DiffMatte (Hu et al., 2024), see Fig 8), which strictly rely on perfect manually annotated trimaps to predict high-quality alpha mattes, our model achieves comparable performance in matting accuracy using only text prompts as guidance, demonstrating better practicality and flexibility of our approach in real-world

applications. We expect that future work will integrate VAE with lossless image compression technology to more effectively address this problem.

Also, inheriting biases from training data may affect our model. For example, the matting model currently used as a tool for labeling GT alphas in RefCOCO may produce labeling errors in some certain situations, or the overall training data may not cover certain visual scenes with extreme lighting or environmental conditions. This will affect the generalizability of our model. We look forward to future developments in referencing image matting datasets that are larger, cover more real-world scenes, and have more accurate annotations to support progress on the soft grounding problem.

## 5 Conclusion

In this paper, we unified various matting tasks as a soft grounding problem, addressing both instance-level localization and transparency estimation using the visual-language capabilities of Stable Diffusion. Instead of directly training SD, we introduced a distillation framework that separates localization and transparency tasks, improving performance in complex scenarios with multiple objects or intricate transparency patterns. This framework also enables a more efficient student model, balancing complexity and accuracy. Extensive evaluations demonstrate our method's effectiveness, with competitive speed and strong generalization across diverse matting categories.

**Broader Impact Statement.** Our method advances text-guided image matting but could be misused for image manipulation or deepfakes, thus we encourage responsible use in creative applications. Besides, since our method leverages pretrained diffusion and vision-language models, it may inherit and amplify biases present in the training data. These biases could manifest as unequal performance across demographic groups or systematic failures in underrepresented categories. Future research should carefully evaluate and document such biases, and explore mitigation strategies such as balanced data curation, prompt robustness analysis, or bias-aware training objectives. Moreover, the performance of our method is inherently sensitive to text prompts, and vague or ambiguous prompts can lead to degraded or unintended results. This raises potential concerns about fairness and reliability when the system is deployed in practical settings. To address this, future work may incorporate adaptive prompting strategies, human-in-the-loop corrections, or uncertainty estimation mechanisms to ensure more robust and transparent usage.

**Acknowledgements.** This research is supported by the Guangdong Natural Science Funds for Distinguished Young Scholars (Grant 2023B1515020097), Natural Science Foundation of Guangdong Province (2023A1515012894), Key R&D Project of Guangzhou Science and Technology Plan (2023B01J0002), National Natural Science Foundation of China (Grant No.: 62502117), the National Research Foundation Singapore under its AI Singapore Programme (AISG Award No: AISG3-GV-2023-011), the National Research Foundation Singapore under the AI Singapore Programme (AISG Award No: AISG4-TC-2025-018-SGKR), the Singapore Ministry of Education AcRF Tier 1 Grant (Grant No.: MSS25C004), and the Lee Kong Chian Fellowships.

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

# 6 Appendix

## 6.1 Analysis of Self-Attention Optimization

Here, we compare the computational overhead of vanilla self-attention (SA) and the proposed sparse self-attention (Sp.SA) within the framework of our method. **Complexity of Vanilla SA.** Let $n$ represent the number of input tokens and $d$ the dimension of the token embedding. The computational complexity of a single SA operation—*i.e.*, the total number of addition and multiplication operations ($\mathcal{O}_{\text{SA}}$)—can be calculated as follows:

$$\mathcal{O}_{\text{SA}} = \underbrace{2nd^2 \times 3}_{Q,\,K,\,V} + \underbrace{2n^2 d}_{Q \cdot K} + \underbrace{n^2}_{(\sqrt{d})^{-1}} + \underbrace{3n^2 - 1}_{\text{Softmax}} + \underbrace{2n^2 d}_{AV} \tag{10}$$
$$= 6nd^2 + 4n^2 d + 4n^2 - 1$$

For instance, given the input feature $f$, computing $Q$ involves multiplying $f \in \mathbb{R}^{n \times d}$ and $W_Q \in \mathbb{R}^{d \times d}$, resulting in $n \times d \times d$ addition operations and $n \times d \times d$ multiplication operations. Hence, the total number for calculating $Q$ is $2nd^2$. The same applies to the computations of $Q \cdot K$ and $AV$. Additionally, since the normalization ($\times (\sqrt{d})^{-1}$) and Softmax operations act on an $n \times n$ matrix, their complexities are $n^2$ and $\underbrace{n^2}_{\text{exponential}} + \underbrace{(n^2 - 1)}_{\text{addition}} + \underbrace{n^2}_{\text{division}} = 3n^2 - 1$, respectively[*].

**Complexity of Optimized SA.** As mentioned in Sec. 3.3 in the main paper, introducing the down-sampled convolution $\phi$ with kernel size and stride both equal to $k \times k$ results in $K^S$ and $V^S$ with a sequence length $k^2$ times shorter than the original $K$ and $V$. Let $j$ ($j = k^2$) be the scale factor of this operation; the corresponding complexity $\mathcal{O}_{\text{Sp.SA}}$ is given as:

---

[*]FLOPs for exponential operation can be seen as 1 due to common instruction optimization and hardware acceleration rather than using repeating multiplication.

$$\mathcal{O}_{\text{Sp.SA}} = \underbrace{2nd^2}_{Q} + \underbrace{2 \times \frac{n}{j} \times d^2 \times 2}_{K^S, V^S \text{ with } \phi \text{ applied}} + \underbrace{2nd \times \frac{n}{j}}_{Q \cdot K^S} + \underbrace{n \times \frac{n}{j}}_{(\sqrt{d})^{-1}}$$
$$+ \underbrace{3n \times \frac{n}{j} - 1}_{\text{Softmax}} + \underbrace{2n \times \frac{n}{j} \times d}_{AV^S} + \underbrace{j \times \frac{n}{j} \times d \times d}_{\text{Convolution operation } \phi} \tag{11}$$
$$= (3 + \frac{4}{j})nd^2 + \frac{4n^2 d}{j} + \frac{4n^2}{j} - 1$$

In this work, we set $k_{32^2} = k_{16^2} = 2$ for SA layers at resolutions of $32 \times 32$ and $16 \times 16$ respectively, with number of channels $d_{32^2} = 640, d_{16^2} = 1280$. Then, the fraction of complexity reduction is calculated as:

$$\frac{\mathcal{O}_{\text{Sp.SA}}(n = 32^2, d = 640, j = 2^2)}{\mathcal{O}_{\text{SA}}(n = 32^2, d = 640)} \approx 0.4515$$
$$\frac{\mathcal{O}_{\text{Sp.SA}}(n = 16^2, d = 1280, j = 2^2)}{\mathcal{O}_{\text{SA}}(n = 16^2, d = 1280)} \approx 0.6176 \tag{12}$$

The results show that the proposed Sp.SA can reduce the computational complexity by 54.85% and 38.24% for the SA layers at $32 \times 32$ and $16 \times 16$ resolution, respectively. Given the same $j$, the complexity reduction is more significant for the SA layer with a longer sequence length, which aligns with the intuition that the computational overhead of SA is proportional to the square of the sequence length.

## 6.2 More Details on Data Curation

### 6.2.1 Annotation Models Choice

For generating caption annotations, we select a common VLM (BLIP-2 (Li et al., 2023b)) to do this. Since captions were generated exclusively for the P3M10K, AM2K, and RM1K datasets, which are trimap-free matting datasets where the most salient object is unambiguously the foreground. Generating captions to describe the foreground in these datasets is a straightforward task for modern VLMs (including BLIP-2). Moreover, the BLIP-2 model was selected for its balance of usability, accuracy, and prevalence in the matting domain. It has been successfully used in prior work (Wang et al., 2024) to generate captions for matting datasets, demonstrating its effectiveness in this context.

For generating GT alpha mattes for the RefCOCO dataset, which contains only segmentation-level annotations, we employed ViTMatte (Yao et al., 2024a) to advance our model's training by producing matting-level annotations. The RefCOCO dataset primarily includes highly opaque objects (e.g., person, animals, tables), where the variance in unknown region of GT mattes is inherently low. Consequently, the choice of matting model has minimal impact on the resulting mattes. ViTMatte was selected for its balance of annotation efficiency and quality.

### 6.2.2 Quality Control

For generated captions, since modern VLMs have achieved superior performance in generating captions for images, the simplicity of this task ensures that errors, if any, are minimal and do not significantly impact training, as the captions only need to capture key foreground characteristics. Consequently, no additional quality control measures were necessary beyond the inherent reliability of these datasets and the robustness of BLIP-2 in this context.

For generated GT alpha mattes, considering certain classes in RefCOCO (e.g., bottles) include both transparent and opaque objects, and their associated prompts often lack explicit transparency indicators (e.g., the word "transparent" is rarely used). To ensure annotation accuracy and avoid confusing the model during training, we excluded samples from these ambiguous classes. Additionally, some classes (e.g., bicycles) have segmentation annotations that lack fine-grained texture details (e.g., spokes in bicycle wheels), leading to

low-quality GT alpha mattes. These samples were also removed to maintain the accuracy of the model's learning process and avoid ambiguity.

### 6.2.3 Data Pre-processing

All the images are resized to $512 \times 512$ for training in all stages. For composition samples from RefMatte (Li et al., 2023a), we use the same augmentation strategy as in Li et al. (2022) to reduce the discrepancy between synthetic and real data.

### 6.3 More Details on Training

**Rationals of Hyperparamter Setting.** For the kernel size of the morphological operation $K_{mor}$, we set it to 15 to balance overestimation (if too large) and underestimation (if too small) of the transparent region, aligning with common practice in matting methods. For the cross-attention loss $\mathcal{L}_{\text{CTM}}$, we use `sum` reduction to address the presence of many zero matrices in the cross-attention maps (e.g., padding tokens), leading to a smaller $\lambda_{\text{CTM}}$ of 0.1. During training, $\mathcal{L}_{\text{STM}}$ and $\mathcal{L}_{\alpha^{lr}}$ are relatively small, while $\mathcal{L}_{\text{SG}}$ is larger under `mean` reduction. To balance these, we set $(\lambda_{\text{STM}}, \lambda_{\alpha^{lr}}, \lambda_{\text{SG}})$ to $(10, 10, 0.5)$.

**Training Hyperparameters.** We show the training parameters in Tab. 4. We adopt AdamW (Loshchilov & Hutter, 2019) as the optimizer with weight decay as 0.01. All the training work is done on NVIDIA A100 80GB GPU(s). For the scheduler, we adopt a LambdaLR scheduler with several specified milestones. When the milestone is not reached, we use the lambda function $f(x) = (1 - \frac{x}{\text{total iterations}})^{0.9}$ to adjust the learning rate. Otherwise, the learning rate is directly adjusted to the preseted value.

Table 4: Hyperparameters for all training stages.

| Training Stage | Initial Learning Rate | Total Iterations | Total Batch Size | Scheduler Value | Scheduler Milestone | GPUs (A100) | Learnable Parameters | Training Time |
|---|---|---|---|---|---|---|---|---|
| Soft Grounding in Semantic | 5e-5 | 50000 | 32 | [0.5, 0.25] | [15000, 40000] | 4 | 861M | 18.5h |
| Soft Grounding in Transparency | 5e-5 | 50000 | 32 | [0.4, 0.25] | [10000, 35000] | 2 | 509M | 20.7h |
| From Soft Grounding to Alpha Matte | 4e-4 | 50000 | 16 | [0.1, 0.05] | [10000, 35000] | 1 | 2.67M | 12.7h |

**Clarification on Baselines Fine-tuning.** Here we provide more clarification regarding the baselines fine-tuning. First, it is well-known that fine-tuning models pretrained on large-scale datasets with small-scale datasets can lead to performance fluctuations (Zhang et al., 2020; Mosbach et al., 2020; Fu et al., 2023), particularly when the training details (e.g., crucial hyperparameters settings and training techniques) during fine-tuning differ significantly from those used in pretraining, or when the model is with high complexity.

For Grounding DINO (Liu et al., 2024b), as its training process is currently close-source, we could only attempt fine-tuning based on the training details provided in its paper. However, we observed that fine-tuning resulted in inferior performance compared to using the pretrained weights. We attribute this to discrepancies in the training approaches comparing with its undisclosed settings.

For PSALM (Zhang et al., 2024), we attempted fine-tuning based on its publicly available training code but found the performance to be unstable and degraded. We believe this is due to the high model complexity (PSALM uses an LLM as its backbone), which amplifies the challenges of fine-tuning on small-scale datasets.

In contrast, for Stable Diffusion (SD) (Rombach et al., 2022), we argue that its advantage lies in the superior interpretability of its internal features and attention maps, which exhibit strong semantic differentiation, as discussed in Sec. 1. This property provides SD with a unique prior advantage when fine-tuned on downstream tasks supported by small-scale datasets, an advantage not shared by LLM-based methods like PSALM. Furthermore, numerous prior works (Zhao et al., 2023a; Lee et al., 2024; Xu et al., 2024a; Guo et al., 2024; Wang et al., 2024) have provided practical support for fine-tuning SD models on various dense prediction tasks, making this approach more mature. Therefore, we believe that SD offers distinct advantages for the soft grounding task.

### 6.4 More about Sparse Up-sampling Module

The design of the sparse up-sampling module is inspired by the progressive refinement decoder proposed in Yu et al. (2021), consisting of two groups of interpolation-convolution groups. All the vanilla convolutional layers are replaced by sparse convolutional layers (Contributors, 2022) in the up-sampling module. The input of each group is the concatenation of the up-interpolated alpha matte and the context feature from the matting decoder (all with a resolution of $512 \times 512$), along with the original image at the target resolution. The sparse convolution layer performs calculations only on areas where the alpha matte value is greater than 0.01 and less than 0.99. To train the module, we attach it to the pre-trained matting decoder and train the entire network using the same loss function as the matting decoder. The learning rate of the module is set to $2e - 4$, and the total number of iterations is set to $10,000$ with a batch size of 16.

### 6.5 More Robust Analysis

Here, we provide more robust analysis of our model given inaccurate prompt and under complex application scenarios.

#### 6.5.1 Quantitative Analysis on Susceptibility of Noisy/Ambiguous Prompt

To quantitatively assess the influence of noisy and ambiguous prompts on our model's performance, we conducted following ablation study via embedding-level disturbance. This study uses the RefMatte-RW100 dataset as the test benchmark. First, for each text prompt in the dataset, we generated two variants version using GPT-4: a noise version and an ambiguous version. Specifically:

- **Noisy Prompt ($\mathcal{T}_{ny}$)**: We instructed GPT-4 to remove all nouns and their corresponding adjective descriptions from the original prompt $\mathcal{T}$, thereby producing a noisy version that lacks key semantic content and contains only useless function words.

- **Ambiguous Prompt ($\mathcal{T}_{amb}$)**: We instructed GPT-4 to retain only nouns related to object categories, discarding all additional descriptive terms, thus creating an ambiguous prompt with reduced contextual specificity.

To precisely quantify the impact of $\mathcal{T}_{ny}$ and $\mathcal{T}_{amb}$ on model performance, we performed measurements at the linguistic embedding level. The linguistic features corresponding to the original prompt ($f_{\mathcal{T}}$), noisy prompt ($f_{\mathcal{T}_{ny}}$), and ambiguous prompt ($f_{\mathcal{T}_{amb}}$) were extracted. We then injected the features of the noisy and ambiguous prompts into the original prompt's feature in a controlled manner, producing interpolated features $\hat{f}_{\mathcal{T}_{ny}}$ and $\hat{f}_{\mathcal{T}_{amb}}$. These interpolated features were used in place of $f_{\mathcal{T}}$ for subsequent inference.

The degree of feature injection was controlled by two interpolation parameters, $\beta_{ny}$ and $\beta_{amb}$, which allowed us to systematically evaluate the impact of noisy and ambiguous prompts. Mathematically, the feature interpolation process is defined as:

$$\hat{f}_{\mathcal{T}_{ny}} = (1 - \beta_{ny})f_{\mathcal{T}} + \beta_{ny}f_{\mathcal{T}_{ny}}, \quad \hat{f}_{\mathcal{T}_{amb}} = (1 - \beta_{amb})f_{\mathcal{T}} + \beta_{amb}f_{\mathcal{T}_{amb}}. \tag{13}$$

Here, $\beta_{ny}, \beta_{amb} \in [0, 1]$ control the extent of noisy or ambiguous feature injection, with $\beta = 0$ corresponding to the original prompt's feature and $\beta = 1$ corresponding to the fully noisy or ambiguous feature.

We tested several values of $\beta_{ny}$ and $\beta_{amb}$ and quantified the experimental results, as shown in Tab. 5.

Our findings indicate that the model's performance is relatively robust to small perturbations from noisy and ambiguous prompts. For both $\beta_{ny}$ and $\beta_{amb}$ at 0.2, the degradation in performance is modest. At $\beta_{ny} = \beta_{amb} = 0.4$, the performance decline becomes more noticeable. However, a significant drop in performance is observed when $\beta_{ny}$ or $\beta_{amb}$ reaches 0.6 or higher. Notably, noisy prompts tend to have a more severe impact on performance compared to ambiguous prompts, particularly at higher $\beta$ values, as evidenced by the larger increases in these metrics.

Table 5: **Ablation Study on Susceptibility of Noisy/Ambiguous Prompt.**

| $\beta_{ny}$ | SAD↓ | MSE↓ | GRAD↓ | CONN↓ |
|---|---|---|---|---|
| 0 | 7.37 | 0.0264 | 6.55 | 5.31 |
| 0.2 | 8.87 | 0.0321 | 7.14 | 6.23 |
| 0.4 | 12.90 | 0.0474 | 9.17 | 7.03 |
| 0.6 | 23.29 | 0.0867 | 18.11 | 12.83 |
| 0.8 | 40.29 | 0.1510 | 31.03 | 19.24 |

(a) Performance with Noisy Prompts ($\beta_{ny}$)

| $\beta_{amb}$ | SAD↓ | MSE↓ | GRAD↓ | CONN↓ |
|---|---|---|---|---|
| 0 | 7.37 | 0.0264 | 6.55 | 5.31 |
| 0.2 | 8.47 | 0.0306 | 6.99 | 5.89 |
| 0.4 | 14.03 | 0.0517 | 9.47 | 6.55 |
| 0.6 | 26.15 | 0.0976 | 19.86 | 13.69 |
| 0.8 | 34.55 | 0.1385 | 23.96 | 16.35 |

(b) Performance with Ambiguous Prompts ($\beta_{amb}$)

### 6.5.2 Qualitative Analysis on More Complex Scenarios

Although our model is susceptible to noisy and ambiguous prompts, it demonstrates strong robustness in more complex scenarios when the prompt is accurate. In Fig. 9 below, we present additional robust qualitative evaluations that clearly support the model's resilience under more challenging conditions, including multiple objects, instruction-containing prompts, and images with more intricate spatial relationships. Furthermore, while using LLM-based paraphrasing for prompt enhancement can partially mitigate this limitation, we believe that a more effective solution lies in incorporating prompt engineering into the network architecture or training process, or exploring adaptive guidance strategies to further improve robustness, which is beyond the scope of this paper. While not with in our scope, addressing this challenge presents an interesting research direction that we leave to future work.

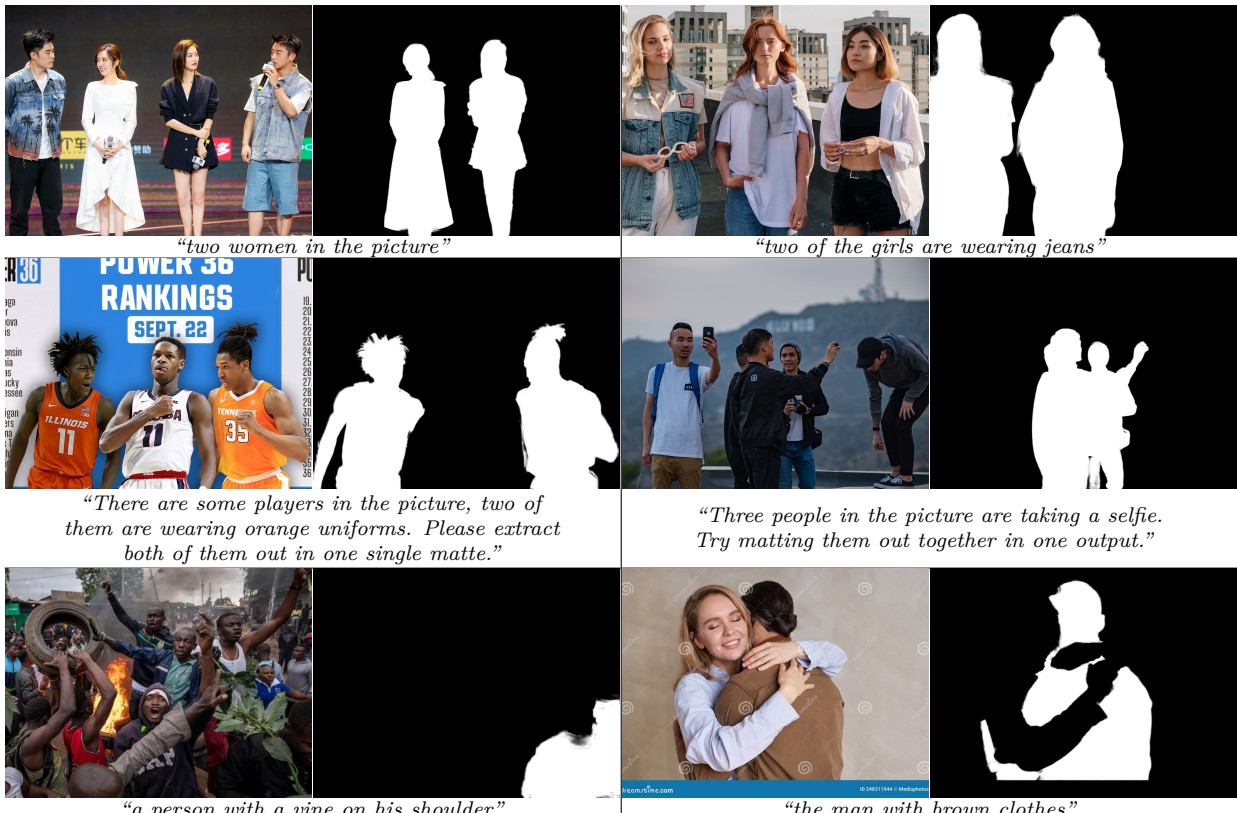

Figure 9: **More Qualitative Robust Evaluation.** Despite some minor prediction artifacts, our model still demonstrates strong robust performance under more complex application scenes.

### 6.6 More Ablation Studies

**Loss Functions.** We also explore the effect of each loss adopted in the distillation process in Tab. 6 using RefMatte-RW100 as the benchmark. Since $\mathcal{L}_{SG}$ is the key loss for semantic guidance from $\epsilon^T\theta$, omitting it will lead to a significant performance drop. $\mathcal{L}_{STM}$ and $\mathcal{L}_{CTM}$ also evidently improve the final performance, demonstrating the effectiveness of transparency exploration and retention of attention information for the optimized $\epsilon_\theta^S$. Furthermore, the learned transparent confidence map $R$ via $\mathcal{L}_R$ also facilitates the matting E-D in predicting more accurate alpha mattes.

Table 6: **Ablation study on different losses.**

| $\mathcal{L}_{\alpha^{lr}}$ | $\mathcal{L}_{SG}$ | $\mathcal{L}_{STM}$ | $\mathcal{L}_{CTM}$ | $\mathcal{L}_R$ | SAD↓ | MSE↓ | GRAD↓ | CONN↓ |
|---|---|---|---|---|---|---|---|---|
| ✓ | | | | | 14.26 | 0.0465 | 8.95 | 7.09 |
| ✓ | ✓ | | | | 8.40 | 0.0297 | 7.25 | 6.11 |
| ✓ | ✓ | ✓ | | | 8.13 | 0.0284 | 7.06 | 5.98 |
| ✓ | ✓ | ✓ | ✓ | | 7.96 | 0.0279 | 6.91 | 5.80 |
| ✓ | ✓ | ✓ | ✓ | ✓ | 7.37 | 0.0264 | 6.55 | 5.31 |

**Caption Annotation Model.** As discussed in Sec. 6.2.1, our training process of our model is not sensitive to the choice of captioning model. To further verify this, we conduct following ablation study. We replace BLIP-2 with Qwen2.5-VL (Bai et al., 2025), a comparable multi-modal LLM, to generate the caption for the training samples without caption annotation. For consistency, we used the same prompt ("*Describe the foreground in the picture, including its clothing, appearance, or behavior.*") for both models. The quantitative results (using RefMatte-RW100 as the benchmark) are presented in Tab. 7. The evaluation metrics show very negligible differences between the two models, indicating that our method is robust to the choice of captioning model.

Table 7: **Ablation study on caption annotation model.**

| Models for Caption Annotation | SAD↓ | MSE↓ | GRAD↓ | CONN↓ |
|---|---|---|---|---|
| BLIP-2 (Li et al., 2023b) | 7.37 | 0.0264 | 6.55 | 5.31 |
| Qwen2.5-VL (Bai et al., 2025) | 7.36 | 0.0261 | 6.56 | 5.29 |

**GT Alpha Matte Annotation Model.** As discussed in Sec. 6.2.1, to annotate GT alpha matte for RefCOCO dataset, the choice of matting model has minimal impact on the resulting alpha mattes. Other matting models could be substituted with similar outcomes. To validate this, we conducted an ablation study by replacing ViTMatte with DiffMatte (Hu et al., 2024) to annotate GT alpha mattes for RefCOCO and retrained stage 2 and stage 3 of our model. We also use RefMatte-RW100 as the benchmark for evaluation. The results, shown in Tab. 8, indicate very negligible performance differences across the evaluation metrics, demonstrating the robustness of our method to the choice of matting model.

Table 8: **Ablation study on GT alpha matte annotation model.**

| Model for GT Matte Annotation | SAD↓ | MSE↓ | GRAD↓ | CONN↓ |
|---|---|---|---|---|
| ViTMatte (Yao et al., 2024a) | 7.37 | 0.0264 | 6.55 | 5.31 |
| DiffMatte (Hu et al., 2024) | 7.37 | 0.0265 | 6.54 | 5.31 |

**Self-Attention Optimization Kernel.** We further evaluate the impact of using different kernel sizes $k$ for the down-sampled convolution $\phi$ in the optimization of SA at various resolutions. Our default setting removes SA layers at a resolution of $64 \times 64$ (denoted as $k_{64^2} = -1$) and sets $k_{32^2} = k_{16^2} = 2$. Therefore, we attempt to increase the kernel size at resolutions of $32 \times 32$ and $16 \times 16$, and also try restoring SA layers at a resolution of $64 \times 64$ with $k_{64^2} = 2$ instead of removing them entirely, resulting in three different settings.

In Tab. 9, we report the quantitative results of these three settings (the last three rows) alongside our default setting (the first row), using RefMatte-RW100 as the benchmark. The inference time per sample is also evaluated, following the same setup as in Sec. 4.3 of the main paper. We found that using a larger kernel size at resolutions of $32 \times 32$ and $16 \times 16$ can lead to performance degradation, as the over-optimized

Table 9: **Ablation study on kernel size $k$ in Self-Attention Optimization.** Here -1 means remove the SA layers directly.

| $k_{64^2}$ | $k_{32^2}$ | $k_{16^2}$ | SAD↓ | MSE↓ | GRAD↓ | CONN↓ | Inference Time (ms) |
|---|---|---|---|---|---|---|---|
| -1 | 2 | 2 | 7.37 | 0.0264 | 6.55 | 5.31 | 95.14 |
| -1 | 2 | 4 | 8.57 | 0.0301 | 7.24 | 6.05 | 94.90 |
| -1 | 4 | 2 | 7.65 | 0.0272 | 6.70 | 5.54 | 94.69 |
| 2 | 2 | 2 | 7.13 | 0.0255 | 6.37 | 5.19 | 130.03 |

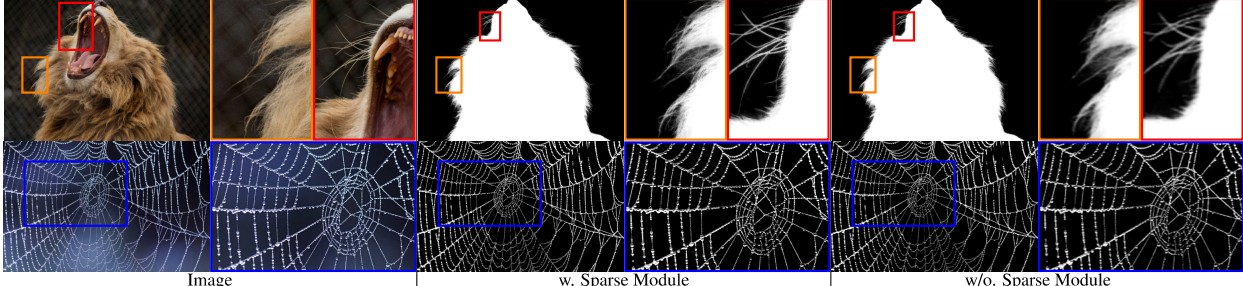

Figure 10: **Qualitative ablation comparison of the sparse up-sampling module (SUM).** The text inputs for the examples from left to right are: 1) *a lion yawning with its mouth open*; 2) *a spider web is shown.* SUM preserves more high-frequency textures, resulting in a clearer alpha matte.

SA layers lose their ability to capture global context and intra-object cohesion information. This effect is particularly noticeable at a resolution of $16 \times 16$, where increasing $k$ causes a significant performance drop. Moreover, the running time is not substantially reduced in either case. We also found that restoring SA layers at a resolution of $64 \times 64$ with $k_{64^2} = 2$ yields only marginal improvements while significantly slowing down the model.

**Sparse Up-sampling Module.** We conduct the ablation study of the sparse up-sampling module on the RefMatte-Test dataset in full resolution, which consists of more natural image matting samples. The quantitative result is shown in Tab. 10. We found that three metrics (SAD, MSE, GRAD) are improved with the sparse up-sampling module, while the CONN metric is slightly higher. This is mainly because an over-smoothed alpha leads to a lower connectivity error value. We then show the qualitative comparison in Fig.10. Despite the higher connectivity error, SUM can produce clearer alpha with finer details, demonstrating the effectiveness of this module in extending our model to arbitrary high-resolution inference, with more accurate matting texture preserved compared with direct up-sampling.

Table 10: **Ablation study on sparse up-sample module (SUM).**

| | SAD↓ | MSE↓ | GRAD↓ | CONN↓ |
|---|---|---|---|---|
| Ours (w/o. SUM) | 24.87 | 0.0114 | 16.43 | 12.58 |
| Ours (w. SUM) | 24.19 | 0.0112 | 14.94 | 12.91 |

**Hyperparameters.** We analyze the impact of hyperparameters in the training process using RefMatte-RW100 as the benchmark (see Tab. 11). We evaluate three alternative settings and find that our default configuration (row 1) consistently outperforms them, demonstrating its effectiveness.

**Stable Diffusion Versions.** To further assess the generalizability of our framework, we apply our method to teach two additional versions of Stable Diffusion (SD v2.0 and SD v2.1) for soft grounding while keeping all other experimental settings unchanged. The evaluation results on RefMatte-RW100 (see Tab. 12) show that compared to our default setting (SD v1.5), SD v2.x achieves better performance, highlighting the broad applicability of our approach.

Moreover, we also notice that recent SD models are emerged for advanced visual generation, such as SDXL (Podell et al., 2023) and SD v3.x series (Esser et al., 2024). SD3.x employs MMDiT, a pure

Table 11: **Ablation study on hyperparameters.**

| $\lambda_{\mathrm{STM}}$ | $\lambda_{\mathrm{CTM}}$ | $\lambda_{\mathrm{SG}}$ | $\lambda_{\alpha^{lr}}$ | $\lambda_R$ | $K_{mor}$ | SAD↓ | MSE↓ | GRAD↓ | CONN↓ |
|---|---|---|---|---|---|---|---|---|---|
| 10 | 0.1 | 0.5 | 10 | 1 | 15 | 7.37 | 0.0264 | 6.55 | 5.31 |
| 10 | 0.1 | 0.5 | 10 | 1 | 5 | 9.32 | 0.0354 | 8.57 | 6.61 |
| 1 | 0.1 | 0.5 | 1 | 1 | 15 | 7.65 | 0.0271 | 6.63 | 5.32 |
| 10 | 0.1 | 1 | 10 | 1 | 15 | 7.41 | 0.0268 | 6.55 | 5.29 |

Table 12: **Ablation study on Stable Diffusion (SD) version.**

| SD version | SAD↓ | MSE↓ | GRAD↓ | CONN↓ |
|---|---|---|---|---|
| SD v1.5 | 7.37 | 0.0264 | 6.55 | 5.31 |
| SD v2.0 | 6.50 | 0.0232 | 6.72 | 4.98 |
| SD v2.1 | 6.18 | 0.0219 | 6.42 | 4.70 |

transformer-based model, as the denoising network, and SDXL incorporates a more complex multi-scale generation structure. Due to these substantial architectural differences from the UNet architecture (comprising residual and attention layers), and given that our method is tailored for UNet-based SD models, we acknowledge that our framework cannot yet be directly applied to these transformer-based SD models.

Nevertheless, we emphasize the advantages of UNet-based SD models for downstream tasks supported by relatively small-scale datasets, such as our soft grounding problem. UNet-based networks benefit from the inductive bias of convolutional layers, which is particularly suited for small datasets and dense prediction tasks like soft grounding. In contrast, transformer-based denoising networks (e.g., MMDiT in SD3) excel in long-term modeling for image generation but rely on large-scale training data and may not be as effective for downstream tasks with limited data. In future work, we plan to explore adapting transformer-based SD models to address our soft grounding problem. However, we also highlight that UNet-based SD models possess unique advantages for transfer to dense prediction tasks.

### 6.7 Towards Video Instance Matting

Leveraging text prior enables us to easily extend our model to video instance matting, since the language is also capable of describing a series of frames. We show the qualitative comparison of our model and several strong baselines on video instance matting in Fig. 11. Without any temporal modeling or fine-tuning, our model can still produce high-quality results on video sample with given text input. Our model is also robust to multi-instance environments and camera viewpoint changes, which demonstrates the powerful practicality of our model in real-world applications. More video instance matting results can also be found in the **supplementary video file**.

### 6.8 Generalization on Non-photographic Data

We further evaluated the generalization ability of our model on several non-photographic images, including animation-style and oil painting-style samples. Although the objects in these samples tend to have clearer boundaries compared to real-world images, they also exhibit markedly different textual characteristics and lighting effects. Since our training data consisted entirely of photographic images, these samples are considered out-of-distribution (OOD) for our model. Nevertheless, as shown in Fig. 12, guided by textual input, our model still demonstrates strong generalization on these samples, highlighting its robustness and practical applicability in more diverse scenarios.

### 6.9 Computational Complexity Comparison

We compare the computational cost of our model with 4 baselines with competitive performance in Tab. 13 with number of parameters, GMac, GFlops, and average GPU memory usage as metrics. Our method has both the lowest model size and computational cost among them, demonstrating the efficiency of ours comprehensively.

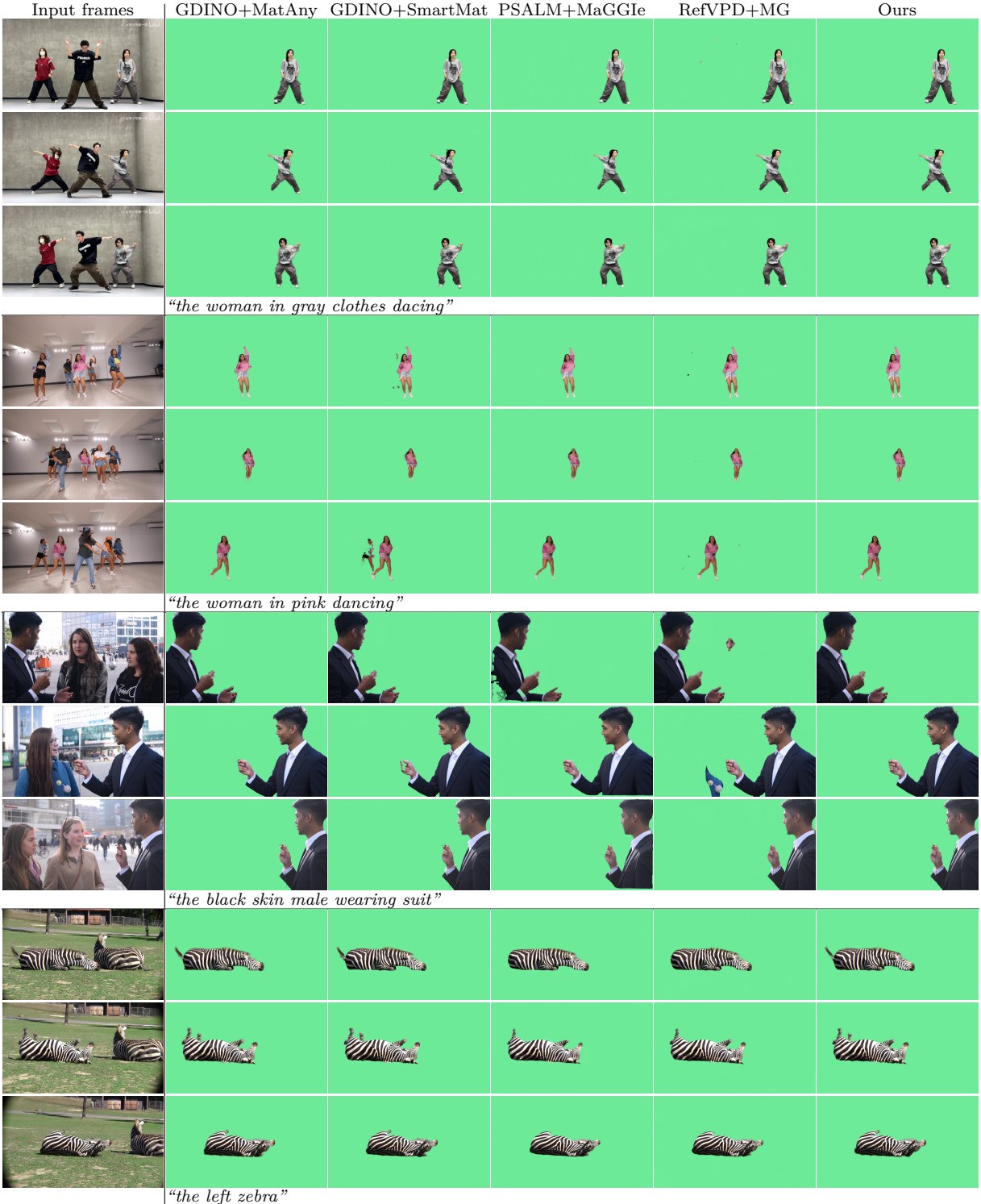

Figure 11: **Qualitative comparison on video instance matting.** The corresponding text inputs are shown in *italic* below every video clip. The frames are placed following the temporal order in the original video from top to down.

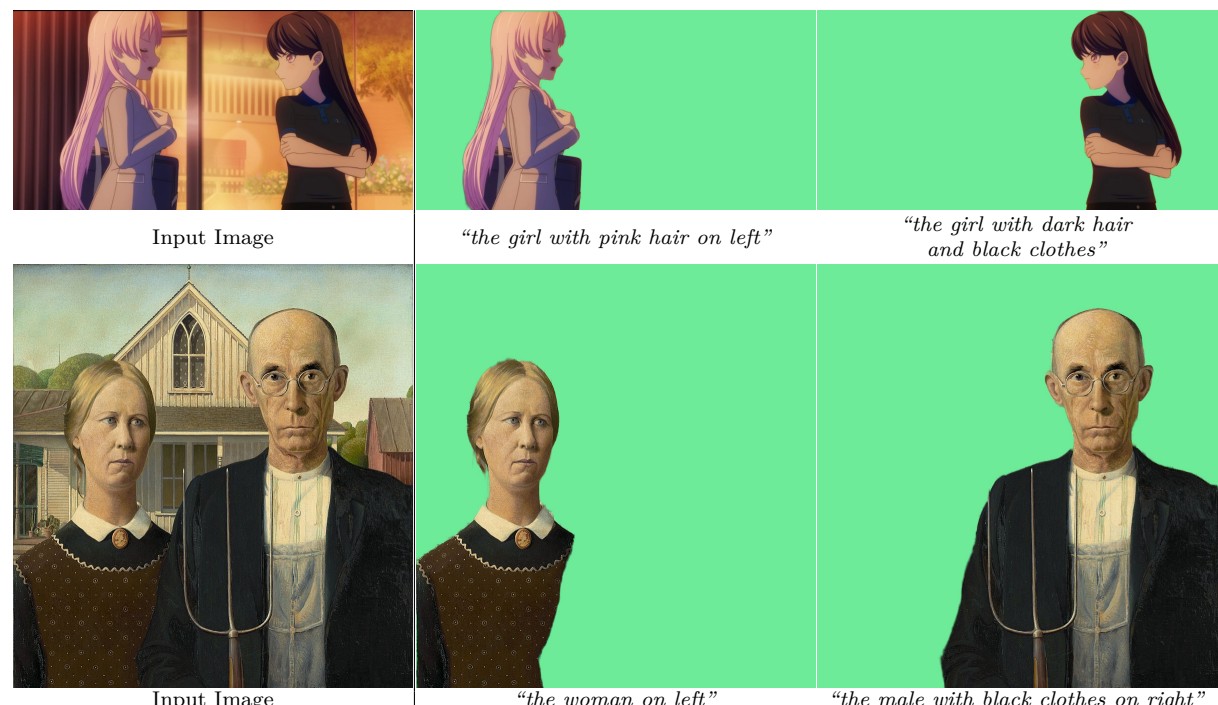

Figure 12: **Qualitative results of our model on several non-photographic images (animation and oil painting).** The corresponding text inputs are shown in *italic* below every sample. Our model shows strong out-of-domain generalizability on such data.

Table 13: **Computational complexity comparison.**

| | Number of parameters (M) ↓ | GMACs↓ | GFLOPs↓ | Average GPU Memory Usage (MiB) ↓ |
|---|---|---|---|---|
| PSALM+MGM | 1617 | 1430 | 2859 | 23372 |
| RefVPD+MGM | 929 | 959.4 | 1919 | 7806 |
| PSALM+MaGGIe | 1617 | 1429 | 2858 | 23236 |
| RefVPD+MaGGIe | 930 | 958.9 | 1918 | 7670 |
| Ours (w/o. optimization) | 861 | 902.3 | 1805 | 7914 |
| Ours | **429** | **769.7** | **1539** | **6474** |

## 6.10 More Qualitative Results

In Fig. 13, we show more qualitative comparison results on RefMatte-Test and RefMatte-RW100. Our model outperform other baselines both on synthetic and real-world datasets, demonstrating the superior performance of our model in grounding alpha matte from text descriptions. We also show more qualitative results on portrait and animal matting (Fig. 14). The text inputs are generated by BLIP2 (Li et al., 2023b). The results further validate the generalization ability of our model on different matting tasks.

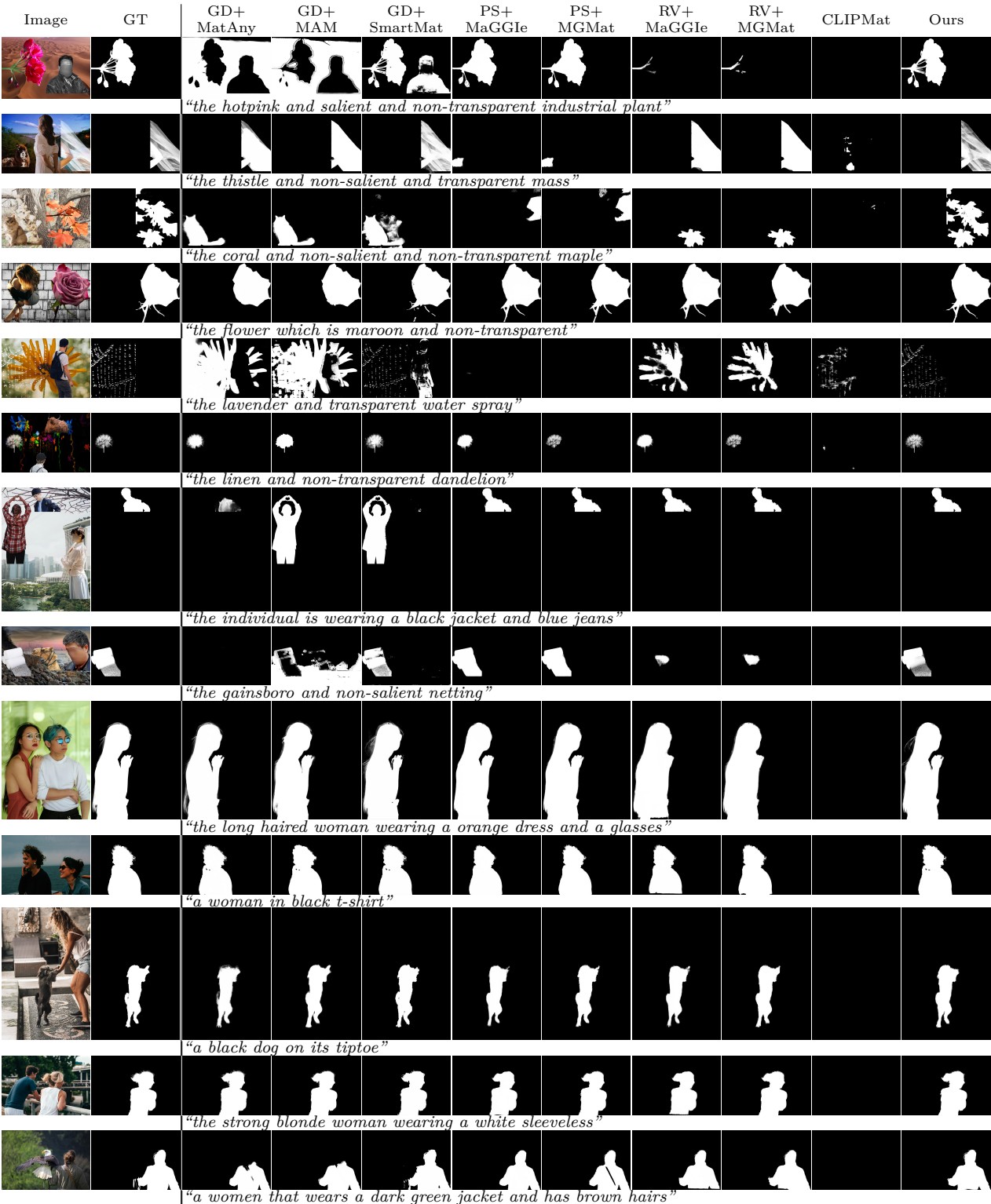

Figure 13: **More qualitative comparisons on soft grounding task.** The text inputs are shown in *italic* below. Zoom in for better view.

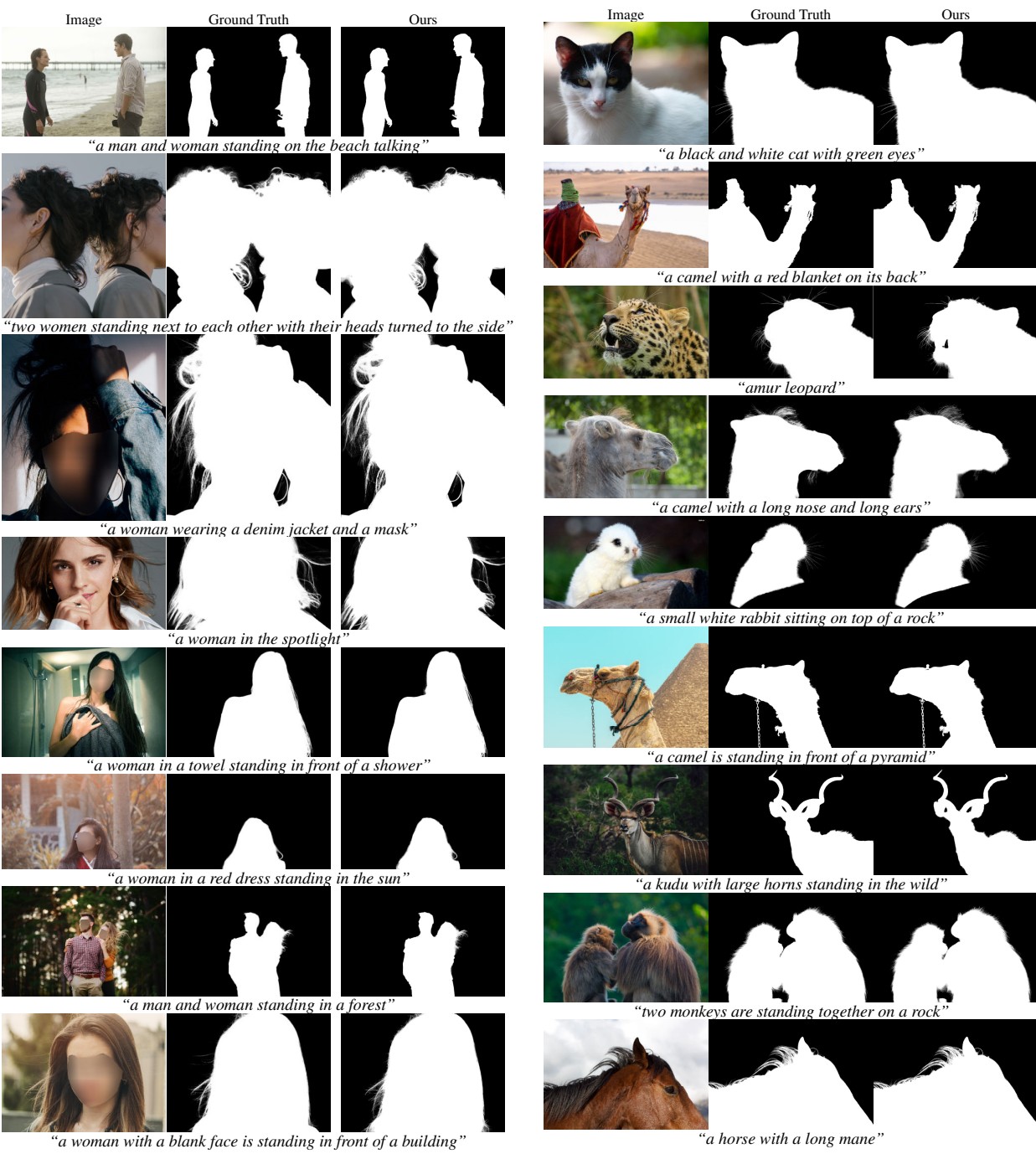

Figure 14: **Qualitative results of our model on portrait (left) and animal (right) matting.** Zoom in for better view.

