# OpenReview forum: "Teaching Diffusion Models to Ground Alpha Matte"
_TMLR — Accepted by TMLR_

### Review · Reviewer_UMCq · 2025-07-10

**Summary Of Contributions:**

The core innovation is a computationally efficient Teacher-Student distillation framework that reframes the task as "soft grounding." The key contributions and new knowledge are:
- State-of-the-Art Performance with High Efficiency: The paper demonstrates that this approach achieves top-tier results across multiple benchmarks. Crucially, it does so while operating with significantly fewer parameters and faster inference speeds than competing methods. This marks a critical step towards the practical deployment of high-quality matting models.
- A Unified Framework for Diverse Matting Tasks: The method successfully subsumes various matting sub-tasks (e.g., portrait, animal, referring matting) under a single, versatile text-promptable model. This demonstrates strong generalization and flexibility, moving the field beyond task-specific architectures toward a more universal solution.

**Audience:**

Yes

**Claims And Evidence:**

No

**Requested Changes:**

To address the concerns raised in the "Weaknesses" section, we propose the following adjustments. We believe these changes are essential for substantiating the paper's core contributions and ensuring its scientific rigor.

1. Clarification of the Data Curation and Annotation Pipeline (Critical)
The paper's reproducibility and the trustworthiness of its results depend heavily on the dataset. The current description is too high-level. We request that the authors add a dedicated subsection, detailing the data generation process. This section should address the following points:
- Quality Control Protocol: Please describe the specific steps taken to validate or filter the auto-generated captions from BLIP-2 and alpha mattes from Yao et al. (2024a). Was there a manual review process? Were metrics used to discard low-quality samples? Providing examples of both accepted and rejected auto-generated labels would be highly beneficial.
- Rationale for Tooling: A justification for choosing BLIP-2 and the specific matting model by Yao et al. (2024a) should be provided. What were their key advantages over other available tools? Please provide the comparison with other annotation models or tools.

2. Data-Level Ablation Study to Isolate Model Contribution (Critical)
The current results confound the contribution of the novel architecture with the contribution of the newly curated dataset. To disentangle these two factors, we request the following experiment:
- Train a Strong Baseline on Your Dataset: Please select a high-performing baseline method from Table 1 (e.g., the previous state-of-the-art) and train it on the exact same training dataset used for your proposed model.
- Report Comparative Results: Add a row to Table 1 or create a new table that compares your method against this retrained baseline.
This experiment is critical because it will provide a fair, apples-to-apples comparison that isolates the architectural improvements. If your model still significantly outperforms the baseline on the same data, it will provide undeniable evidence that your proposed teacher-student framework is the primary driver of the performance gains, thereby solidifying the paper's core contribution. Without this, it remains unclear whether the novelty lies in the model or the data.

**Strengths And Weaknesses:**

Strengths
- Clarity and Motivation: The paper is exceptionally well-written and clearly articulated. The authors provide strong intuition and compelling motivation for their core design choices, especially the teacher-student distillation framework for disentangling localization from transparency estimation. The logic is easy to follow and methodologically sound.
- Compelling Empirical Performance: The method achieves state-of-the-art results, demonstrating significant improvements over a comprehensive set of baselines on multiple benchmarks, as evidenced by the results in Table 1. The performance gains are not marginal but substantial, establishing a new benchmark for text-driven matting.
- Effective Visualizations: The submission is supported by high-quality, informative visualizations. For example, Figure 3 offers an excellent and intuitive diagram of the entire pipeline, which greatly aids in understanding the complex interactions between the teacher, student, and various loss components.

Weaknesses
- Lack of Detail in Data Curation: The methodology for creating the training dataset lacks transparency. The authors state they use BLIP-2 for caption generation and an existing matting model (Yao et al., 2024a) for alpha mask generation. However, this process raises several critical questions that are not addressed:
  - Quality Control: What measures were implemented to ensure the quality of the automatically generated labels? Noisy or inaccurate captions/mattes could significantly impact training, and the filtering or verification process is not described.
  - Rationale for Model Choice: The paper does not justify why these specific models were chosen for annotation over other alternatives.
  - Performance Sensitivity: An analysis of how the final model's performance is affected by the quality of these auto-generated labels is missing. This is a crucial detail for reproducibility and understanding the robustness of the method.
- Absence of Data-Level Ablation Studies: The paper presents a thorough ablation study on its architectural components (Table 3) but critically omits any ablation on the data itself. This makes it difficult to disentangle the source of the impressive performance gains. The key unanswered question is: How much of the improvement comes from the novel architecture versus the newly curated training data? To isolate the contribution of the proposed model, it would be essential to conduct experiments where a strong baseline method is also trained on the exact same dataset. Without this comparison, the core architectural contribution is confounded with the data's contribution.

---

> ### Author Response · Authors · 2025-08-25
> **Reply to Reviewer UMCq (1/2)**
>
> **W1** & **RC1**: *Lack of Detail in Data Curation*
> **A**: Thanks for your feedback. Below, we address the concerns raised about the quality control measures, rationale for model selection, and performance sensitivity analysis for the automatically generated captions and GT alpha mattes.
> ### Adopt BLIP-2 for caption generation:
>  - **Model Choice and Sensitivity**: As described in Sec. 4.1 of our paper, captions were generated exclusively for the P3M10K, AM2K, and RM1K datasets, which are trimap-free matting datasets where the most salient object is unambiguously the foreground. Generating captions to describe the foreground in these datasets is a straightforward task for modern VLMs (including BLIP-2). Thus, our training process of our model is not sensitive to the choice of captioning model. The BLIP-2 model was selected for its balance of usability, accuracy, and prevalence in the matting domain. It has been successfully used in prior work [1] to generate captions for matting datasets, demonstrating its effectiveness in this context. To further demonstrate that our model’s training is not sensitive to the choice of captioning model, we conduct following ablation study. We replace BLIP-2 with Qwen2.5-VL [2], a comparable multi-modal LLM, to generate the caption for the training samples without caption annotation. For consistency, we used the same prompt (`Describe the foreground in the picture, including its clothing, appearance, or behavior.`) for both models. The quantitative results (using RefMatte-RW100 as the benchmark) are presented in the table below. The evaluation metrics show very negligible differences between the two models, indicating that our method is robust to the choice of captioning model.
>
> | Model for Caption Annotation | SAD   | MSE   | Grad  | Conn  |
> |------------|-------|-------|-------|-------|
> | BLIP-2     | 7.37  | 0.0264| 6.55  | 5.31  |
> | Qwen2.5-VL | 7.36  | 0.0261| 6.56  | 5.29 |
>
>  - **Quality Control**: As we have mentioned above, modern VLMs have achieved superior performance in generating captions for images. The simplicity of this task ensures that errors, if any, are minimal and do not significantly impact training, as the captions only need to capture key foreground characteristics. Consequently, no additional quality control measures were necessary beyond the inherent reliability of these datasets and the robustness of BLIP-2 in this context.
>
> ### Adopt ViTMatte (Yao et al., 2024a) [3] for GT matte generation:
>  - **Model Choice and Sensitivity**: For generating GT alpha mattes for the RefCOCO dataset, which contains only segmentation-level annotations, we employed ViTMatte [3] to advance our model’s training by producing matting-level annotations. The RefCOCO dataset primarily includes highly opaque objects (e.g., person, animals, tables), where the variance in unknown region of GT alpha mattes is inherently low. Consequently, the choice of matting model has minimal impact on the resulting alpha mattes. ViTMatte was selected for its balance of annotation efficiency and quality, but other matting models could be substituted with similar outcomes. To validate this, we conducted an ablation study by replacing ViTMatte with DiffMatte [4] to annotate GT alpha mattes for RefCOCO and retrained stages 2–3 of our model. We also use RefMatte-RW100 as the benchmark for evaluation. The results, shown in the table below, indicate very negligible performance differences across the evaluation metrics, demonstrating the robustness of our method to the choice of matting model.
>
> | Model for GT Matte Annotation | SAD   | MSE   | Grad  | Conn  |
> |------------|-------|-------|-------|-------|
> | ViTMatte (Yao et al., 2024a)  | 7.37  | 0.0264| 6.55  | 5.31  |
> | DiffMatte                     | 7.37  | 0.0265| 6.54  | 5.31  |
>
>  - **Quality Control**: To generate pseudo-trimaps from segmentation annotations in RefCOCO, we applied morphological operations, which are effective for fully opaque objects. However, certain classes in RefCOCO (e.g., bottles) include both transparent and opaque objects, and their associated prompts often lack explicit transparency indicators (e.g., the word `transparent` is rarely used). To ensure annotation accuracy and avoid confusing the model during training, we excluded samples from these ambiguous classes. Additionally, some classes (e.g., bicycles) have segmentation annotations that lack fine-grained texture details (e.g., spokes in bicycle wheels), leading to low-quality GT alpha mattes. These samples were also removed to maintain the accuracy of the model’s learning process and avoid ambiguity.

---

> > ### Author Response · Authors · 2025-08-25
> > **Reply to Reviewer UMCq (2/2)**
> >
> > (Continued)
> >
> > While we acknowledge that there is room for enhancement, particularly in quality control measures, we would like to emphasize that the primary contribution of this study lies in developing a novel algorithm to enable diffusion models to address a new task in multi-modal matting. Our experiments based on current training data, as detailed in the paper, sufficiently demonstrate the advantages of the proposed algorithm. In future work, we plan to construct a larger-scale, higher-quality multi-modal matting dataset to further advance the field.
> >
> > We have already included the clarification on model choice and analysis on quality control in Sec. 6.2 of the revised manuscript. The ablation studies on sensitivity are also added in Sec. 6.6 in the revision.
> >
> > **W2** & **RC2**: *Absence of Data-Level Ablation Studies:*
> > **A**: Thanks for your comment. To address this concern, we refer the reviewer to the second paragraph of Sec. 4.3 in our paper, where we have already clarified the fairness ensurement related to the same training data for further performance comparison. Specifically, two strong baseline models (RefVPD+MaGGIe and RefVPD+MGMat) were trained on the exact same dataset as our proposed method. Additionally, the CLIPMat model was also trained on the identical dataset. Our experimental results (Tab. 1) demonstrate that, despite aligning the training data across all baseline modules, our proposed algorithm achieves significant performance improvement. This clearly underscores the importance of our core architectural contributions.
> >
> > To further clarify this point, we have revised the corresponding analysis part in the manuscript to explicitly highlight that the observed performance gains primarily stem from our novel soft grounding problem modeling and the carefully designed distillation framework, rather than the newly curated dataset.
> >
> >
> > ### Reference
> > [1] Wang et al.: Matting by generation. In: SIGGRAPH Conference Papers, pp. 1–11 (2024)
> >
> > [2] Bai et al.: Qwen2.5-VL technical report. arXiv preprint arXiv:2502.13923 (2025)
> >
> > [3] Yao et al.: Vitmatte: Boosting image matting with pre-trained plain vision transformers. In: Information Fusion, vol. 103, pp. 102091 (2024)
> >
> > [4] Hu et al.: Diffusion for natural image matting. In: ECCV, pp. 181–199 (2024)

---

> > > ### Comment · Reviewer_UMCq · 2025-08-28
> > >
> > > Thanks for your response. It addresses some of my concerns.

---

### Review · Reviewer_AoAt · 2025-08-11

**Summary Of Contributions:**

This paper considers image matting problem, which aims to estimate the per-pixel transparency (i.e., alpha matte) to separate foreground and background, by leveraging pre-trained text-to-image diffusion models (e.g., Stable Diffusion). Unlike traditional methods that rely on visual priors such as trimaps or masks, which are often expensive to obtain, there is growing research that uses text-based vision models such as CLIP (i.e., CLIPMat) or Stable Diffusion. However, the author claims that those approaches also lack pixel-level precision or struggles with complex transparency and multi-object cases. To this end, the authors propose an alternative approach to achieve high-quality image matting using diffusion models. In specific, the authors frame matting problem as a soft grounding problem, where it jointly solves instance-level localization and transparency estimation using textual information. They introduce two-stage teacher-student distillation framework, where 1) teacher model is trained by adapting Stable Diffusion for soft semantic grounding (only localization), trained to classify each pixel into foreground, background, or transparent regions using cross-entropy loss with CLIP text embeddings. Then 2), the student model is a lightweight, structure-optimized version distilled from the teacher to perform transparency estimation while preserving semantic localization. Here, they introduce semantic guidance loss that conducts feature-level distillation from teacher, and transparency mining loss that uses self and cross attention maps from teacher to mine transparency cues. Lastly, the matting decoder is attached to refine a coarse alpha matte and a transparency confidence map into full-resolution output.

Through evaluation on RefMatte-Testset, RefMatte-RW100, AM2K (animal), and P3M-P (portrait), the authors show that the proposed method outperforms CLIPMat and cascaded baselines (e.g., GroundingDINO + matting or PSALM + matting) in both accuracy and speed. Furthermore, it demonstrates zero-shot generalization capability among diverse matting tasks without tuning.

**Audience:**

Yes

**Broader Impact Concerns:**

The paper could discuss broader impact concerns in their paper. For example, the proposed method could be misused for image manipulation or deepfakes as they provide high-quality matting of natural images. Furthermore, the biases from pretrained model could be inherited to their model, thus analyzing those biases in depth would be helpful.

**Claims And Evidence:**

Yes

**Requested Changes:**

### Detailed analysis on prompt dependency.
Since the method relies on precise, unambiguous text prompts, there should be more robustness analysis rather than few qualitative examples. For example, the authors could add experiments on how noisy or ambiguous prompts affect quantitative performance. Here, I believe the method could be further improved by using prompt enhancement techniques (e.g., using LLM-based paraphrasing). The related question is, if the prompt becomes more detailed, could the model handle more complex scenarios (e.g., multi-object, complex descriptions or complex spatial relationships)?

### Real-world applications.
The proposed evaluation is limited to static image benchmarks, but does the method can be applied to dynamic cases (e.g., video matting)? If true, showing video matting experiments would further improve the paper’s contribution. Also, to show out-of-domain generalization, it could be possible to show generalization on non-photographic images such as cartoons or synthetic images.

### Comparisons with baselines.
While PSALM, Grounding DINO + matting could be a firm baseline, they are used with their original weights. Thus, it remains a question why fine-tuning such models degrade the performance, while using Stable Diffusion + proposed method works. Also, showing the parameter counts, FLOPs, or memory usage to contextualize speed vs accuracy trade-offs would strengthen the paper.

### Ablation studies on backbone.
The proposed method uses Stable Diffusion v2.0 and v2.1, but it could be interesting if the method applies to different models. For example, how does the proposed method behave when using older models (e.g., SDv1.5) or recent models (e.g., SDXL, SD3, SD3.5)? The scaling behavior of the backbone size would be more beneficial for future research.

### Miscellaneous.
While I believe the paper is clearly written, the presentation could be improved. For instance, Figure 3 is too dense to understand the major detail at glance, and the equations to explain the loss function may overwhelm the readers. Simplifying the diagram for method description and equations would help the readers to understand paper more easily.

**Strengths And Weaknesses:**

### Strengths
The proposed approach seems sound that unifies diverse matting tasks under the soft grounding task by integrating localization and transparency in a single model. Also, the proposed teacher-student design seems to work well as it disentangles the localization and transparency learning through leveraging the visual priors of pretrained text-to-image diffusion models. It also enjoys computational efficiency compared to baselines through structure pruning and sparse attention that reduce inference time and supports high-resolution inference efficiently. Lastly, the generalization capability demonstrates the practical usage of the proposed method.


### Weaknesses
One potential problem would be the dependency towards text prompts, where the models’ performance would vary among the descriptiveness of prompt inputs, and ambiguous text prompts would limit its performance. Also, while the teacher-student framework seems sound, it requires multiple stages in training (i.e., semantic grounding - distillation - matting refinement) with multiple datasets and text annotations, which makes the training complicated. The proposed method would heavily depend on the quality of the pretrained diffusion model, thus the diffusion backbone’s biases would potentially be inherited. The sparse attention enjoys efficient inference, but it seems like there is a slight degradation in performance compared to the unoptimized student model.

---

> ### Author Response · Authors · 2025-08-25
> **Reply to Reviewer AoAt (1/5)**
>
> **W1**: *One potential problem would be the dependency towards text prompts, where the models’ performance would vary among the descriptiveness of prompt inputs, and ambiguous text prompts would limit its performance.*
> **A**: Thank you for your feedback. As discussed in Sec. 4.6 of our manuscript, the performance of prompt-based systems is indeed influenced by the quality and descriptiveness of the input prompts. This characteristic is not unique to our method but is a well-known aspect of vision-language models, as noted in prior literatures [1-4]. Our approach assumes reasonably constructed prompts, consistent with established practices in the field. Addressing challenges related to ambiguous or low-quality prompts falls within the broader scope of prompt engineering, which is beyond the focus of this study. However, we acknowledge this as a valuable direction for future research. To clarify this point, we have incorporated the above discussion into Sec. 4.6 of the revised manuscript.
>
> **W2**: *Also, while the teacher-student framework seems sound, it requires multiple stages in training (i.e., semantic grounding - distillation - matting refinement) with multiple datasets and text annotations, which makes the training complicated.*
> **A**: Thank you for your comment. We would like to clarify that this design reflects a deliberate trade-off between all-in-one model training and a modular, multi-stage approach that disentangles sub-tasks. Neither approach is inherently superior, as each presents distinct advantages and challenges. Directly fine-tuning an all-in-one model may simplify the training pipeline but increases optimization complexity, as it requires the model to address a target task that significantly diverges from the pre-training objective. In contrast, our modular framework disentangles semantic localization and transparency estimation—two sub-tasks with conflicting requirements, as the former relies on high-level semantics and the latter on low-level details. This design enables more effective supervision, improved interpretability, and enhanced robustness. As demonstrated in Tab. 3, our disentangled approach outperforms direct fine-tuning and single-stage models in both performance and efficiency. Additionally, the refinement stage significantly improves accuracy by enhancing the coarse alpha output.
>
> **W3**: *The proposed method would heavily depend on the quality of the pretrained diffusion model, thus the diffusion backbone’s biases would potentially be inherited.*
> **A**: Thank you for your feedback. We acknowledge that our method, like other approaches leveraging pretrained backbones, may inherit biases from the diffusion model. However, **the focus of our work is NOT on perfecting the diffusion backbone but on effectively adapting it to perform a novel task**, which represents a significant contribution to the field. It is well-established that vision-language models pretrained on large-scale datasets, such as diffusion model series, consistently outperform those trained on task-specific datasets (e.g., matting-only datasets) due to their superior generalization and broader applicability. Our work demonstrates how diffusion models can be effectively tailored for a targeted application, advancing the state-of-the-art in this domain. Additionally, as shown in Tab. 11 of the manuscript, we have already conducted ablation studies across different versions of Stable Diffusion (SD). Our method imposes no constraints on U-Net initialization, and the results demonstrate robustness across various U-Net-based diffusion backbones.
>
> Moreover, we recognize that inheriting biases from pretrained Stable Diffusion models is an inherent limitation of our approach. These biases may impose an upper bound on our model's ability to interpret complex prompt content, as this capability is constrained by the pretrained model's understanding of such prompts. Additionally, biases may adversely affect performance in handling images with challenging conditions, such as those with highly complex lighting or shadows, potentially limiting the general applicability of our model. To clarify this concern transparently, we have included the above discussion in Sec. 4.6 of the revised manuscript.

---

> > ### Author Response · Authors · 2025-08-25
> > **Reply to Reviewer AoAt (2/5)**
> >
> > **W4**: *The sparse attention enjoys efficient inference, but it seems like there is a slight degradation in performance compared to the unoptimized student model.*
> > **A**: Thank you for the comment. We acknowledge that the structural optimization will introduce a slight performance degradation compared to the unoptimized model. However, it is important to note that our optimized model still outperforms most baseline methods, as demonstrated in our evaluations (see Tab. 1), validating the effectiveness of our framework in modeling soft grounding. Furthermore, the optimized student model achieves a significant 33.46% reduction in inference time (from 143ms to 95.14ms), as discussed in Sec. 4.5. This reduction is particularly valuable in practical applications, such as mobile devices, where lower computational requirements enhance user experience. In future work, we aim to explore more advanced optimization strategies to achieve real-time matting while further minimizing performance degradation. We have incorporated this discussion into Sec. 4.5 of the revised manuscript to better highlight the trade-offs and practical benefits of our approach.

---

> > > ### Author Response · Authors · 2025-08-25
> > > **Reply to Reviewer AoAt (3/5)**
> > >
> > > **RC1**: *Detailed analysis on prompt dependency.*
> > > **A**: Thank you for your valuable feedback. To quantitatively assess the influence of noisy and ambiguous prompts on our model’s performance, we conducted following ablation study via embedding-level disturbance. This study uses the RefMatte-RW100 dataset as the test benchmark. First, for each text prompt in the dataset, we generated two variants version using GPT-4: a **noisy** version and an **ambiguous** version. Specifically:
> > >
> > > - **Noisy Prompt ($\mathcal{T} _ {ny}$)**: We instructed GPT-4 to remove all nouns and their corresponding adjective descriptions from the original prompt $\mathcal{T}$, thereby producing a noisy version that lacks key semantic content and contains only useless function words.
> > > - **Ambiguous Prompt ($\mathcal{T} _ {amb}$)**: We instructed GPT-4 to retain only nouns related to object categories, discarding all additional descriptive terms, thus creating an ambiguous prompt with reduced contextual specificity.
> > >
> > > To precisely quantify the impact of $\mathcal{T} _ {ny}$ and $\mathcal{T} _ {amb}$ on model performance, we performed measurements at the linguistic embedding level. The linguistic features corresponding to the original prompt ($f _ {\mathcal{T}}$), noisy prompt ($f _ {\mathcal{T} _ {ny}}$), and ambiguous prompt ($f _ {\mathcal{T} _ {amb}}$) were extracted. We then injected the features of the noisy and ambiguous prompts into the original prompt’s feature in a controlled manner, producing interpolated features $\hat{f} _ {\mathcal{T} _ {ny}}$ and $\hat{f} _ {\mathcal{T} _ {amb}}$. These interpolated features were used in place of $f _ {\mathcal{T}}$ for subsequent inference.
> > >
> > > The degree of feature injection was controlled by two interpolation parameters, $\beta _ {ny}$ and $\beta _ {amb}$, which allowed us to systematically evaluate the impact of noisy and ambiguous prompts. Mathematically, the feature interpolation process is defined as:
> > >
> > > $
> > > \hat{f} _ {\mathcal{T} _ {ny}} = (1 - \beta _ {ny}) f _ {\mathcal{T}} + \beta _ {ny} f _ {\mathcal{T} _ {ny}},
> > > $
> > >
> > > $
> > > \hat{f} _ {\mathcal{T} _ {amb}} = (1 - \beta _ {amb}) f _ {\mathcal{T}} + \beta _ {amb} f _ {\mathcal{T} _ {amb}}.
> > > $
> > >
> > > Here, $\beta _ {ny}, \beta _ {amb} \in [0, 1]$ control the extent of noisy or ambiguous feature injection, with $\beta = 0$ corresponding to the original prompt’s feature and $\beta = 1$ corresponding to the fully noisy or ambiguous feature.
> > >
> > > We tested several values of $\beta _ {ny}$ and $\beta _ {amb}$ and quantified the experimental results, as shown in the tables below:
> > >
> > > | $\beta _ {ny}$ | SAD  | MSE  | Grad | Conn |
> > > |----------------|------|------|------|------|
> > > | 0            | 7.37  | 0.0264| 6.55  | 5.31 |
> > > | 0.2            | 8.87 | 0.0321 | 7.14| 6.23 |
> > > | 0.4            | 12.90 | 0.0474 | 9.17 | 7.03 |
> > > | 0.6            | 23.29 | 0.0867 | 18.11 | 12.83 |
> > > | 0.8            | 40.29 | 0.1510 | 31.03 | 19.24 |
> > >
> > > | $\beta _ {amb}$ | SAD  | MSE  | Grad | Conn |
> > > |-----------------|------|------|------|------|
> > > | 0             | 7.37  | 0.0264| 6.55  | 5.31 |
> > > | 0.2             | 8.47 | 0.0306 | 6.99 | 5.89 |
> > > | 0.4             | 14.03 | 0.0517 | 9.47 | 6.55 |
> > > | 0.6             | 26.15 | 0.0976 | 19.86 | 13.69 |
> > > | 0.8             | 34.55 | 0.1385 | 23.96 | 16.35 |
> > >
> > > Our findings indicate that the model's performance is relatively robust to small perturbations from noisy and ambiguous prompts. For both $\beta _ {ny}$ and $\beta _ {amb}$ at $0.2$, the degradation in performance is modest. At $\beta _ {ny} = \beta_{amb} = 0.4$, the performance decline becomes more noticeable. However, a significant drop in performance is observed when $\beta _ {ny}$ or $\beta _ {amb}$ reaches $0.6$ or higher. Notably, noisy prompts tend to have a more severe impact on performance compared to ambiguous prompts, particularly at higher $\beta$ values, as evidenced by the larger increases in these metrics.
> > >
> > > Although our model is susceptible to noisy and ambiguous prompts, it demonstrates strong robustness in more complex scenarios when the prompt is accurate. In Fig. 8 of the revised manuscript, we present additional robust qualitative evaluations that clearly support the model’s resilience under more challenging conditions, including multiple objects, instruction-containing prompts, and images with more intricate spatial relationships. Furthermore, while using LLM-based paraphrasing for prompt enhancement can partially mitigate this limitation, we believe that a more effective solution lies in **incorporating prompt engineering into the network architecture or training process**, or **exploring adaptive guidance strategies to further improve robustness**, which is beyond the scope of this paper. While not with in our scope, addressing this challenge presents an interesting research direction that we leave to future work. The above experiments and discussions have been added to Sec. 6.5 of the revised manuscript.

---

> > > > ### Author Response · Authors · 2025-08-25
> > > > **Reply to Reviewer AoAt (4/5)**
> > > >
> > > > **RC2**: *Real-world applications.*
> > > > **A**: Thank you for your feedback. We have provided results of our model in dynamic cases within the video file (**video_result.mp4**) included in the Supplementary Material. Without any temporal modeling or fine-tuning, our model demonstrates stable performance. Additionally, we have showcased the out-of-distribution (OOD) generalization capabilities of our model on animation and oil painting samples in Sec. 6.8 and Fig. 11 of the revised manuscript. Although objects in these samples tend to have clearer boundaries compared to real-world images, they also exhibit markedly different textual characteristics and lighting effects compared to photographic images. Nevertheless, guided by textual input, our model demonstrates strong generalization on these samples, highlighting its robustness and practical applicability in diverse scenarios.
> > > >
> > > > **RC3**: *Comparisons with baselines.*
> > > > **A**: Thank you for your insightful comments. It is well-known that fine-tuning models pretrained on large-scale datasets with small-scale datasets can lead to performance fluctuations [5-7], particularly when the training details (e.g., crucial hyperparameters settings and training techniques) during fine-tuning differ significantly from those used in pretraining, or when the model is with high complexity. For Grounding DINO, as its training process is currently close-source, we could only attempt fine-tuning based on the training details provided in its paper. However, we observed that fine-tuning resulted in inferior performance compared to using the pretrained weights. We attribute this to discrepancies in the training approaches comparing with its undisclosed settings. For PSALM, we attempted fine-tuning based on its publicly available training code but found the performance to be unstable and degraded. We believe this is due to the high model complexity (PSALM uses an LLM as its backbone), which amplifies the challenges of fine-tuning on small-scale datasets. In contrast, for Stable Diffusion (SD), we argue that its advantage lies in the superior interpretability of its internal features and attention maps, which exhibit strong semantic differentiation, as discussed in Sec. 1 of our paper. This property provides SD with a unique prior advantage when fine-tuned on downstream tasks supported by small-scale datasets, an advantage not shared by LLM-based methods like PSALM. Furthermore, numerous prior works [8-12] have provided practical support for fine-tuning SD models on various dense prediction tasks, making this approach more mature. Therefore, we believe that SD offers distinct advantages for the soft grounding task. These discussions have been incorporated into Sec. 6.3 of the revised manuscript.
> > > >
> > > > Additionally, we have added measurements of Average GPU Memory Usage and reported the computational complexity of our unoptimized model in Tab. 13 of the revised manuscript to better contextualize the trade-offs between complexity and performance.

---

> > > > > ### Author Response · Authors · 2025-08-25
> > > > > **Reply to Reviewer AoAt (5/5)**
> > > > >
> > > > > **RC4**: *Ablation Studies on Backbone*
> > > > > **A**: Thank you for your suggestion. We clarify that the default backbone for our method is Stable Diffusion (SD) v1.5, and all experiments in this paper were conducted using a backbone initialized with SD v1.5 weights. Additionally, we have already demonstrated the effectiveness of our algorithm in eliminating backbone initialization bias by presenting results with different SD versions (SD v2.0/2.1) for backbone initialization in the last subsection of Sec. 6.6 and Tab. 12 of the paper.
> > > > >
> > > > > Regarding recent models (e.g., SDXL, SD3, SD3.5), we note that these models are not simple parameter scale-ups from SD 2.x. SD3.x employs MMDiT, a pure transformer-based model as the denoising network, which significantly differs in structure from the UNet architecture (comprising residual and attention layers). Similarly, SDXL incorporates a more complex multi-scale generation structure. Due to these substantial architectural differences, and given that our method is tailored for UNet-based SD models, we acknowledge that it cannot yet be directly applied to transformer-based SD models.
> > > > >
> > > > > Nevertheless, we emphasize the advantages of UNet-based SD models for downstream tasks supported by relatively small-scale datasets, such as our soft grounding problem. UNet-based networks benefit from the inductive bias of convolutional layers, which is particularly suited for small datasets and dense prediction tasks like soft grounding. In contrast, transformer-based denoising networks (e.g., MMDiT in SD3) excel in long-term modeling for image generation but rely on large-scale training data and may not be as effective for downstream tasks with limited data. In future work, we plan to explore adapting transformer-based SD models to address our soft grounding problem. However, we also highlight that UNet-based SD models possess unique advantages for transfer to dense prediction tasks.
> > > > >
> > > > > We have added the above discussion to Sec. 6.6 of the revised manuscript for clearer clarification.
> > > > >
> > > > > **RC5**: *Miscellaneous*
> > > > > **A**: Thank you for your constructive feedback. In the revised manuscript, we have updated Fig. 3 by removing certain lines to simplify the model diagram. Additionally, we have revised Eq. 3 to improve its clarity and conciseness.
> > > > >
> > > > > **Broader Impact Concerns**: *The paper could discuss broader impact concerns in their paper. For example, the proposed method could be misused for image manipulation or deepfakes as they provide high-quality matting of natural images. Furthermore, the biases from pretrained model could be inherited to their model, thus analyzing those biases in depth would be helpful.*
> > > > > **A**: We have added a broader impact statement in Sec. 5 of the revised manuscript to discuss potential ethical implications related to the misusing of our method.
> > > > >
> > > > > ### Reference
> > > > > [1] Zhang, Y., Fan, C.-K., Huang, T., Lu, M., Yu, S., Pan, J., Cheng, K., She, Q., Zhang, S.: AutoV: Learning to Retrieve Visual Prompt for Large Vision-Language Models. arXiv preprint arXiv:2506.16112 (2025)
> > > > > [2] Wu, S., Sun, M., Wang, W., Wang, Y., Liu, J.: VisualPrompter: Prompt Optimization with Visual Feedback for Text-to-Image Synthesis. arXiv preprint arXiv:2506.23138 (2025)
> > > > > [3] Qu, X., Gou, G., Zhuang, J., Yu, J., Song, K., Wang, Q., Li, Y., Xiong, G.: Proapo: Progressively automatic prompt optimization for visual classification. In: CVPR, pp. 25145–25155 (2025)
> > > > > [4] Wen, Z., Weng, L., Tang, Y., Zhang, R., Liu, Y., Pan, B., Zhu, M., Chen, W.: Exploring multimodal prompt for visualization authoring with large language models. arXiv preprint arXiv:2504.13700 (2025)
> > > > > [5] Zhang, T., Wu, F., Katiyar, A., et al.: Revisiting few-sample BERT fine-tuning. arXiv preprint arXiv:2006.05987 (2020)
> > > > > [6] Mosbach, M., Andriushchenko, M., Klakow, D.: On the stability of fine-tuning BERT: Misconceptions, explanations, and strong baselines. arXiv preprint arXiv:2006.04884 (2020)
> > > > > [7] Fu, Z., So, A. M.-C., Collier, N.: A stability analysis of fine-tuning a pre-trained model. arXiv preprint arXiv:2301.09820 (2023)
> > > > > [8] Zhao, W., Rao, Y., Liu, Z., et al.: Unleashing text-to-image diffusion models for visual perception. In: ICCV, pp. 5729–5739 (2023)
> > > > > [9] Lee, H.-Y., Tseng, H.-Y., Yang, M.-H.: Exploiting diffusion prior for generalizable dense prediction. In: CVPR, pp. 7861–7871 (2024)
> > > > > [10] Xu, G., Ge, Y., Liu, M., et al.: Diffusion models trained with large data are transferable visual models. arXiv preprint arXiv:2403.06090 (2024)
> > > > > [11] Guo, H., Ye, Z., Cao, Z., et al.: In-context matting. In: CVPR, pp. 3711–3720 (2024)
> > > > > [12] Wang, Z., Li, B., Wang, J., et al.: Matting by generation. In: SIGGRAPH, pp. 1–11 (2024)

---

### Review · Reviewer_SfhP · 2025-08-12

**Summary Of Contributions:**

The paper proposes a unified formulation of diverse image matting tasks as a soft grounding problem, simultaneously addressing instance-level localization and transparency estimation guided by natural language prompts.

The authors introduce a two-stage teacher–student distillation framework, where a Stable Diffusion model is first adapted for semantic localization, and its knowledge is transferred to a lightweight student model for transparency prediction. The designs include 1) Novel loss designs: including a Semantic Guidance Loss for feature-level alignment and 2) Transparency Mining Losses leveraging attention maps: enabling effective disentanglement of the two objectives. Structural optimizations such as block pruning and sparse self-attention substantially improve inference efficiency, and a sparse upsampling module allows high-resolution inference while preserving fine details.

**Audience:**

Yes

**Broader Impact Concerns:**

The authors should add a Broader Impact Statement for discussion of possible ethical implications, such as the mitigation of dataset bias and prompt sensitivity, training data bias, etc.

**Claims And Evidence:**

No

**Requested Changes:**

I am not an expert in image matting, but I know pretty well about the diffusion-based generation. According to the weaknesses discussed before, my main concerns are as follows:

1. While introducing the prompt guidance seems reasonable in Fig. 1, the author should provide more motivations on tuning a diffusion model, particularly for generation, into a matting network. From my understanding, some designs for image generation in stable diffusion models are unfriendly for matting tasks. For example, the downsampling of VAE will lead to information loss and affect the accuracy of matting, which requires high alignment in pixel space. Furthermore, whether the matting network requires such a heavy network is another problem. I do not think matting relies on the generation ability of the model, which may even counteract the accuracy of the pixel-level alignment of the matting results.

2. The author should provide more details on how to tune the stable diffusion (SD) model following Fig. 3, i.e., how to turn the multi-step SD into a single-step matting network.

3. The author should provide more insights on adopting the Two-Stage Teacher–Student Distillation. It is unclear why distillation is a better option compared with directly tuning the SD model.

4. The author should provide more details on the baseline settings. Specifically, it is unclear whether the compared baselines in the experiments are retrained under the same setting. For example, MatAnyone and Maggie are trained for human matting and may not be suitable for natural images.

5. The author may consider comparing with common matting baselines on non-referring commonly used benchmarks. I regard this as a possible option since the proposed method can assign the target for matting, thus it should be able to handle non-referring data as previous methods, e.g., [Lin et al., 2021].

6. The author may explain more on the potential bias of the training data, especially for the alpha annotations for RefCOCO, which are generated via morphological operations + another matting model (Yao et al., 2024a). The ground truth might carry the bias of that model.

7. For image generation like SD, generating accurate content following the prompt is still an unsolved problem. The author may consider discussing the potential problem of the prompt following in this matting task.

8. The paper provides the video results in Fig. 9 of the supplementary material. It seems the result on human matting, e.g., 'the woman in pink dance' is not very good with many incorrect matting details. The author should provide other video matting results such as GD+MatAny.

[Lin et al., 2021] Real-Time High-Resolution Background Matting

**Strengths And Weaknesses:**

Strengths:

- The paper introduces a new reasonable unified formulation for image matting, i.e., proposing to reformulate diverse matting tasks (portrait, natural image, animal, video, etc.) into a single soft grounding framework, where both instance-level localization and pixel-level transparency estimation are guided by natural language descriptions.
- The paper proposes a successful practice to finetune a diffusion model, i.e., stable diffusion (SD) into a matting network with specific designs, including a Two-Stage Teacher–Student Distillation Framework and Designs for Disentanglement.

Weaknesses:

- The necessity of tuning a diffusion model, particularly for generation, into a matting network is in doubt. From my understanding, some designs for image generation in stable diffusion models are unfriendly for matting tasks. For example, the downsampling of VAE will lead to information loss and affect the accuracy of matting, which requires high alignment in pixel space. Furthermore, whether the matting network requires such a heavy network is another problem. I do not think matting relies on the generation ability of the model, which may even counteract the accuracy of the pixel-level alignment of the matting results.
- The training process of tuning the SD network into a matting network is kind of vague. It is unclear for me how the author turns the multi-step SD into a single-step matting network.
- The motivation of adopting the Two-Stage Teacher–Student Distillation requires more explanation. It is unclear why distillation is a better option compared with directly tuning the SD model.
- The baseline settings need more details. Specifically, it is unclear whether the compared baselines in the experiments are retrained under the same setting. For example, MatAnyone and Maggie are trained for human matting and may not be suitable for natural images. Moreover, the author may consider comparing with common matting baselines on non-referring commonly used benchmarks. I regard this as a possible option since the proposed method can assign the target for matting, thus it should be able to handle non-referring data as previous methods, e.g., [Lin et al., 2021].
- Alpha annotations for RefCOCO are generated via morphological operations + another matting model (Yao et al., 2024a), so the ground truth might carry the bias of that model.

[Lin et al., 2021] Real-Time High-Resolution Background Matting

---

> ### Author Response · Authors · 2025-08-25
> **Reply to Reviewer SfhP (1/4)**
>
> **W1.1** & **RC1.1**: *Necessity of tuning a generation-aimed diffusion model.*
> **A**: Thank you for the insightful comments. The primary objective of our work is to address the challenge of extracting precise alpha mattes for objects guided by text prompts, with the critical sub-problem being instance-level localization through establishing pixel-level cross-modal correspondence between vision and language. Previous approaches, such as our baseline CLIPMat, attempt to tackle this sub-problem by leveraging pretrained CLIP models. However, the cross-modal priors in pretrained CLIP models are inherently limited to image-level understanding, which poses a fundamental challenge in learning more precise and complex pixel-level cross-modal correspondences. In contrast, our proposed algorithm builds upon the Stable Diffusion (SD) model, not for its generative capabilities, but for its rich, robust internal features and highly interpretable cross-attention maps. As discussed in Sec. 1, paragraph 3 of the main paper and illustrated in Fig. 2, despite being designed for generation tasks, the SD model demonstrates strong semantic differentiation in its features and cross-modal attention maps, exhibiting both inter-correlation and intra-consistency. The visualization results in Fig. 2 further highlight that the SD model’s pixel-level cross-modal priors significantly outperform those of CLIP-based models. Additionally, many prior works[1-9] have successfully leveraged SD’s robust pixel-level cross-modal priors for dense visual tasks, reinforcing its suitability for addressing problems like soft grounding, as tackled in this work. Our experimental results in Tab. 1 of the main paper also demonstrate substantial performance improvements over CLIP-based methods.
> To provide further clarity, we have added a detailed discussion of our motivation for using the SD model in Sec. 1 of the revised manuscript.
>
> **W1.2** & **RC1.2**: *Some designs for image generation in SD models are unfriendly for matting tasks. For example, the downsampling of VAE will lead to information loss and affect the accuracy of matting, which requires high alignment in pixel space.*
> **A**: While some aspects of the SD model, such as the downsampling in its variational autoencoder (VAE), may lead to accuracy loss in matting tasks, we emphasize that our approach does not solely rely on fine-tuning the SD model. Instead, we introduce tailored designs to address the high pixel-level accuracy requirements of matting tasks. Specifically, our Stage 3 employs lightweight modules to refine coarse matting results from the SD model back to the original resolution, effectively mitigating detail loss caused by the SD architecture. The ablation study in Tab. 3 (row 5 vs. row 6) of the main paper validates that Stage 3 significantly enhances accuracy by refining coarse alpha outputs. Furthermore, we propose a sparse upsampling module to further refine and upsample matting results to arbitrary resolutions below 2K, meeting the precision demands of most matting applications. The quantitative and qualitative ablation studies in Fig. 9 and Tab. 10 of the main paper clearly demonstrate the effectiveness of this module. In summary, we have incorporated several task-specific designs to adapt the SD model for matting, ensuring both high accuracy and practical applicability.
>
> **W1.3** & **RC1.3**: *Whether the matting network requires such a heavy network?*
> **A**: It is reasonable that our network is heavier than those used in simpler matting scenarios, as our task is fundamentally more complex and general. Traditional matting methods often assume auxiliary inputs such as trimaps or known backgrounds, and typically focus on a single, salient object. In contrast, we tackle the general text-to-matting problem, which requires text-guided instance-level localization without such assumptions. This setting involves handling multiple similar objects and complex textures, making it natural to incorporate a more expressive, data-driven prior like Stable Diffusion (SD), which is well-suited for such general and unconstrained grounding tasks.
>
> To mitigate the added model complexity, we introduce several optimization strategies, including layer pruning and attention refinement, which significantly improve computational efficiency while preserving performance, as demonstrated in Tab. 1. We also recognize the potential of emerging lightweight vision-language models to further improve soft grounding, and we consider this a promising direction. A detailed explanation of our motivation for using SD has been added to Sec. 1 of the revised manuscript.

---

> > ### Author Response · Authors · 2025-08-25
> > **Reply to Reviewer SfhP (2/4)**
> >
> > **W2** & **RC2**: *It is unclear for me how the author turns the multi-step SD into a single-step matting network.*
> > **A**: Thank you for your constructive feedback. The technique of tuning a multi-step SD model into a single-step prediction network has been well-established in prior works[4,5,10]. In our approach, we transform the multi-step SD model into a deterministic single-step perception model by using the image latent without added noise as the input to the denoising U-Net, with the timestep set to $1.0$ during both training and inference.
> > This configuration is consistent with previous works[4,5,10] and effectively adapts a multi-step SD model originally designed for generation into a single-step model suitable for perceptual tasks.
> > We have included a detailed clarification of these training details in Sec. 4.1 of the revised manuscript.
> >
> > **W3** & **RC3**: *More insights on adopting the Two-Stage Teacher–Student Distillation.*
> > **A**: Thank you for your valuable comment. Our two-stage teacher–student distillation framework serves two critical purposes: (1) disentangling the sub-tasks to enable more effective learning, and (2) balancing performance and computational efficiency during inference. The inherent challenges of soft grounding mainly involve two conflicting sub-tasks: semantic localization and transparency estimation. Semantic localization relies on high-level semantic understanding, whereas transparency estimation depends on fine-grained, low-level details. Jointly optimizing these sub-tasks within a single model often results in performance trade-offs due to their competing objectives.
> > In Stage 1, the teacher model focuses on the first sub-task, localizing each component of the alpha matte. Continuous fine-tuning after Stage 1 risks degrading the teacher’s semantic capacity, which our approach avoids. Instead of continuing to fine-tune the teacher model, Stage 2 employs a student U-Net to facilitate effective joint learning of both sub-tasks. As demonstrated in Tab. 3 of the main paper, this disentangled design outperforms both direct fine-tuning and single-stage models in terms of accuracy and efficiency.
> > We have incorporated these insights into the revised manuscript in Sec. 1 and Sec. 3.2.2 to clarify the motivation and benefits of our distillation framework.
> >
> > **W4** & **RC4**: *The baseline settings need more details. Specifically, it is unclear whether the compared baselines in the experiments are retrained under the same setting. For example, MatAnyone and Maggie are trained for human matting and may not be suitable for natural images.*
> > **A**: Thank you for your constructive feedback regarding the baseline settings. As discussed in Sec. 4.3 of the original manuscript, all baselines for the matting component were aligned through fine-tuning on the same training data. Since the training code for the relevant matting networks is publicly available, we utilized their official implementations as the basis for fine-tuning these models in our experiments.
> > To clarify, we did not include MatAnyone as a baseline for the matting component. The baselines evaluated include MatAny, MAM, SmartMat, MGMatting, and MaGGIe. Among these, only MaGGIe was trained for human matting exclusively, while the others natively support natural image matting. Although MaGGIe is trained for human matting, its network architecture does not incorporate specific optimizations tailored exclusively for human subjects. Furthermore, the authors of MaGGIe demonstrated its strong zero-shot generalization capabilities on non-human matting tasks in their paper, indicating that MaGGIe can adapt to natural image matting to some extent.
> > While there may be an adaptation gap for MaGGIe, the other baselines, which natively support natural image matting, were fine-tuned on the same dataset as our model. As shown in Tab. 1 of the original manuscript, these baselines still exhibit inferior performance compared to our proposed model, sufficiently demonstrating the superior performance of our approach. We have added these clarifications to Sec. 4.3 of the revised manuscript to provide a clearer description of the baseline settings and their relevance.

---

> > > ### Author Response · Authors · 2025-08-25
> > > **Reply to Reviewer SfhP (3/4)**
> > >
> > > **W4** & **RC5**: *Moreover, the author may consider comparing with common matting baselines on non-referring commonly used benchmarks. I regard this as a possible option since the proposed method can assign the target for matting, thus it should be able to handle non-referring data as previous methods, e.g., [Lin et al., 2021].*
> > > **A**: Thank you for your valuable comment. To demonstrate the generalization ability of our model, we have included comparisons with several task-specific matting baselines on two non-referring commonly used benchmarks, AM2K and P3M-P, as presented in Sec. 4.4, Tab. 2, and Fig. 5 of the original manuscript. For all test samples in these benchmarks, we fixed the prompt input as `the foreground person` for the P3M-P dataset and `the foreground animal` for the AM2K dataset. This ensures that the input conditions for our model and the baselines are consistent, enabling a fair comparison.
> > > Our findings indicate that while baseline models excel in their specialized matting tasks, they often lack generalization capability and perform poorly in referring image matting tasks. In contrast, our model leverages textual prior knowledge to effectively generalize across various matting tasks, although it may slightly underperform task-specific experts in their respective domains. Most importantly, our model demonstrates superior performance in referring image matting tasks, where other baseline models exhibit suboptimal results.
> > > Regarding the BGMv2 model（Lin et al., 2021）[11] you have mentioned, we note that it requires not only the input image but also a background image that is pixel-level aligned with the input image perfectly. BGMv2 heavily relies on visual background priors provided by this additional background image to perform matting, which fundamentally differs from our model’s reliance on language-based text guidance. In addition, BGMv2 cannot accurately control the extraction of one exact instance in a multi-instance environment, which is very different from our settings. Due to these significant discrepancy discussed, a direct comparison with BGMv2 would be unfair. Consequently, we did not include BGMv2 as a baseline in the experiments presented in Sec. 4.4.
> > >
> > > **W5** & **RC6**: *Potential bias of the training data.*
> > > **A**: Thank you for the constructive suggestions. Actually, the reviewer UMCq has raised a very similar question, and we have already made a detailed reply there. Here we summarize：the RefCOCO dataset primarily includes highly opaque objects (e.g., person, animals, tables), where the variance in unknown region of GT alpha mattes is inherently low. Consequently, the choice of matting model has minimal impact on the resulting alpha mattes. ViTMatte (Yao et al., 2024a) [12] was selected for its balance of annotation efficiency and quality, but other matting models could be substituted with similar outcomes. To validate this, we conducted an ablation study by replacing ViTMatte with DiffMatte [13] to annotate GT alpha mattes for RefCOCO and retrained stages 2–3 of our model. We also use RefMatte-RW100 as the benchmark for evaluation. The results, shown in the table below, indicate very negligible performance differences across the evaluation metrics, demonstrating the robustness of our method to the choice of matting model.
> > > | Model for GT Matte Annotation | SAD   | MSE   | Grad  | Conn  |
> > > |------------|-------|-------|-------|-------|
> > > | ViTMatte (Yao et al., 2024a)  | 7.37  | 0.0264| 6.55  | 5.31  |
> > > | DiffMatte                     | 7.37  | 0.0265| 6.54  | 5.31  |
> > >
> > > The above ablation study and discussion have been added to Sec. 6.6 of the revised manuscript.
> > >
> > > Furthermore, we acknowledge that our model may be affected by potential biases in training data. For example, the matting model currently used as a tool for labeling GT alphas in RefCOCO may produce labeling errors in some certain situations, or the overall training data may not cover certain visual scenes with extreme lighting or environmental conditions. This will affect the generalizability of our model. We look forward to future developments in referencing image matting datasets that are larger, cover more real-world scenes, and have more accurate annotations to support progress on the soft grounding problem. To clarify this concern transparently, we have included the above discussion in Sec. 4.6 of the revised manuscript.

---

> > > > ### Author Response · Authors · 2025-08-25
> > > > **Reply to Reviewer SfhP (4/4)**
> > > >
> > > > **RC7**: *For image generation like SD, generating accurate content following the prompt is still an unsolved problem. The author may consider discussing the potential problem of the prompt following in this matting task.*
> > > > **A**: We acknowledge that our model, which builds upon a pretrained SD model, may face limitations in accurately interpreting prompts under certain challenging conditions. Specifically, the performance of our method is constrained by the inherent prompt understanding capabilities of the pretrained SD model. For instance, SD models often struggle with concepts requiring precise counting, such as prompts like `the eleventh person from the left`, leading to difficulties in accurately interpreting such positional information. These limitations stem from the SD model itself.
> > > > However, we emphasize that the primary focus of our work is **NOT to perfect the diffusion backbone but to effectively adapt it for a novel task**. This distinction underscores the unique and significant contributions of our approach in leveraging pretrained models for targeted applications. We also note that future advancements in vision-language models with improved prompt understanding capabilities could further enhance progress in soft grounding tasks, and we look forward to such developments. The corresponding clarification has been added in Sec. 4.6 of the revised manuscript, discussing the impact of prompt-following challenges on our method.
> > > >
> > > > **RC8**: *The paper provides the video results in Fig. 9 of the supplementary material. It seems the result on human matting, e.g., 'the woman in pink dance' is not very good with many incorrect matting details. The author should provide other video matting results such as GD+MatAny.*
> > > > **A**: Thank you for your valuable feedback. In response, we have updated Fig. 10 in the revised manuscript to include comparisons with some strong baselines, such as GD+MatAny, for video matting tasks. Our method demonstrates superior overall performance, producing more accurate alpha mattes compared to these baselines.
> > > > While the matting result for the example `the woman in pink dance` exhibits some artifacts at the edges of the foreground, we attribute this to the reduced image quality caused by the relatively lower resolution of the original video. Notably, the artifacts are also present in the results of other baselines, often to a more severe degree. In contrast, for cases with higher original video quality, such as the example `the black-skinned male wearing a suit`, our model consistently generates high-quality matting results. These additional results in Fig. 10 further highlight the robustness and effectiveness of our approach.
> > > >
> > > > **Broader Impact Concerns**: *The authors should add a Broader Impact Statement for discussion of possible ethical implications, such as the mitigation of dataset bias and prompt sensitivity, training data bias, etc.*
> > > > **A**: We have added a broader impact statement in Sec. 5 of the revised manuscript to discuss potential ethical implications.
> > > >
> > > >
> > > >
> > > > ### Reference
> > > > [1] Tian, J., Aggarwal, L., Colaco, A., et al.: Diffuse attend and segment: Unsupervised zero-shot segmentation using stable diffusion. In: CVPR, pp. 3554–3563 (2024)
> > > > [2] Wang, J., Li, X., Zhang, J., et al.: Diffusion model is secretly a training-free open vocabulary semantic segmenter. IEEE Transactions on Image Processing (2025)
> > > > [3] Karazija, L., Laina, I., Vedaldi, A., et al.: Diffusion models for zero-shot open-vocabulary segmentation. arXiv preprint arXiv:2306 (2023)
> > > > [4] Zhao, W., Rao, Y., Liu, Z., et al.: Unleashing text-to-image diffusion models for visual perception. In: ICCV, pp. 5729–5739 (2023)
> > > > [5] Lee, H.-Y., Tseng, H.-Y., Yang, M.-H.: Exploiting diffusion prior for generalizable dense prediction. In: CVPR, pp. 7861–7871 (2024)
> > > > [6] Corradini, B. T., Shukor, M., Couairon, P., et al.: Freeseg-diff: Training-free open-vocabulary segmentation with diffusion models. arXiv preprint arXiv:2403.20105 (2024)
> > > > [7] Guo, H., Ye, Z., Cao, Z., et al.: In-context matting. In: CVPR, pp. 3711–3720 (2024)
> > > > [8] Burgert, R., Ranasinghe, K., Li, X., et al.: Peekaboo: Text to image diffusion models are zero-shot segmentors. arXiv preprint arXiv:2211.13224 (2022)
> > > > [9] Xu, J., Liu, S., Vahdat, A., et al.: Open-vocabulary panoptic segmentation with text-to-image diffusion models. In: CVPR, pp. 2955–2966 (2023)
> > > > [10] Xu, G., Ge, Y., Liu, M., et al.: Diffusion models trained with large data are transferable visual models. arXiv preprint arXiv:2403.06090 (2024)
> > > > [11] Lin, S., Ryabtsev, A., Sengupta, S., et al.: Real-time high-resolution background matting. In: CVPR, pp. 8762–8771 (2021)
> > > > [12] Yao, J., et al.: Vitmatte: Boosting image matting with pre-trained plain vision transformers. In: Information Fusion, vol. 103, pp. 102091 (2024)
> > > > [13] Hu, Y., et al.: Diffusion for natural image matting. In: ECCV, pp. 181–199 (2024)

---

> > > > > ### Comment · Reviewer_SfhP · 2025-08-31
> > > > >
> > > > > Thanks for the response of the author.
> > > > > Most of my concerns have been addressed.
> > > > >
> > > > > I still suggest that the author should include the discussion of ``Some designs for image generation in SD models are unfriendly for matting tasks.`` in the revision, e.g., the limitation section. While I understand that most of existing popular open-sourced SDs are latent based and it could be tought for the author to verify the accuracy loss led by the latent space, the loss caused by such paradigm seems hard to be avoided and I still think the accuracy is important for the matting task.

---

> > > > > > ### Author Response · Authors · 2025-09-01
> > > > > >
> > > > > > Thank you for responding to our comments. We are pleased to see that we have addressed most of your concerns. Regarding your suggestion, we acknowledge that some structures in SD designed specifically for generation tasks (such as down-sampling in VAE) may cause accuracy loss in matting tasks, which demand high pixel-level precision. To mitigate this, we have introduced Stage 3 in our framework for detail recovering, as we have discussed for W1.2. Moreover, it is worth emphasizing that our approach primarily leverages the rich semantic prior knowledge encoded in SD models. Consequently, a trade-off between semantic accuracy and fine-grained detail accuracy is normal and reasonable. Despite SD being designed for generation tasks, our tailored designs, particularly Stage 3, enable our model to achieve superior overall matting accuracy compared to baselines, as demonstrated in Tab. 1 of the main paper. We anticipate that future advancements, such as integrating VAEs with lossless image compression techniques, could further alleviate accuracy limitations in matting tasks, and we consider this a promising direction for future work.
> > > > > > We have incorporated the above discussion in the revised manuscript, specifically in the limitations section.

---

### Decision · Action_Editor_v4gu · 2025-09-10

**Recommendation:** Accept with minor revision

**Audience:**

Yes

**Audience Explanation:**

This paper presents an interesting approach to harness knowledge learned in generative diffusion models for a different task, supported by sufficient empirical results. The AE believes the idea and method may be of particular interest to a sub-community of TMLR focused on image generation and the joint modeling of generation and understanding in the image space.

**Claims And Evidence:**

Yes

**Claims Explanation:**

This paper makes two primary claims:

* It introduces a new technical approach to distill a Stable Diffusion model for alpha matte prediction by designing tailored learning objectives, thereby enabling the training of a student model for more efficient inference.

* Empirical experiments demonstrate that the proposed method achieves competitive quality and speed, outperforming baselines on several soft grounding tasks.

During the rebuttal, several concerns were adequately resolved, including dependencies on text prompts, the motivation for the approach, and details of data curation. The authors also clarified design choices and supplemented comparisons on quality and efficiency, which strengthened the overall verification of their claims. Following these responses, all three reviewers recommended leaning towards acceptance. No major outstanding concerns remain, apart from two issues: (1) the explanation of latent diffusion versus pixel-space diffusion, particularly in light of the coarse-grained results yielded by the VAE in latent diffusion, and (2) the extent to which the superior performance gains are attributable to the large parameter capacity of the underlying Stable Diffusion model, as noted by Reviewer SfhP.

The AE agrees with the unanimous reviewer assessment and recommends acceptance, contingent on the authors addressing these remaining concerns in the final version. Specifically, the AE expect the following changes which include
1. A discussion and in-depth analysis of the latent vs. pixel diffusion for the task; for example by analyzing the pixel-level accuracy of the failure cases to show the limit of the latent diffusion.
1. Clarify to what extent the improvements stem from the proposed distillation and learning objectives rather than simply from inheriting the SD model. For example, one can switch a weaker backbone to analyze the performance differences.

---

> ### Author Response · Authors · 2025-10-10
>
> Dear Associate Editor,
>
> Thank you for your response and valuable feedback. Here we show our responses to the minor concerns below and summarize the changes made (marked in **bold**) in the camera-ready version of our paper.
>
> **Q1**: *In-depth analysis of latent vs. pixel diffusion for the task.*
> **A**: We acknowledge that pixel-level diffusion can achieve superior detail accuracy in matting tasks, such as DiffMatte [1]. However, it is important to note that current pixel-level diffusion-based matting models heavily rely on finely crafted manual trimaps as semantic guidance. When trimaps lack sufficient precision (e.g., those generated from segmentation maps), the detail accuracy of such models degrades significantly. To verify this, we have included a qualitative comparison in **Fig. 8 of the revised manuscript**. Given the same image input, we provide DiffMatte with two types of trimaps: (1) a perfect manual trimap and (2) a coarse trimap generated from the segmentation map. We find that DiffMatte produces alpha mattes with better detail accuracy when using the manual trimap. However, when guided by the coarse trimap, DiffMatte fails to produce satisfactory results, especially in the unknown regions (e.g., hair). In contrast, our model generates high-quality alpha mattes with rich details using only the prompt as guidance, despite some minor detail loss. This demonstrates the better practicality and flexibility of our approach in real-world applications compared to pixel-level diffusion methods.
> The corresponding discussion has been added to **Sec. 4.6 of the revised manuscript**.
>
> **Q2**: *Clarify to what extent the improvements stem from the proposed distillation and learning objectives rather than simply from inheriting the SD model.*
> **A**: We emphasize that our ablation studies (Sec. 4.5 and Tab. 3) have already demonstrated the contribution of our proposed algorithm to the overall performance. Directly fine-tuning the SD model without modifications (Tab. 3, row 1) yields suboptimal results. In contrast, our complete algorithmic framework (Tab. 3, row 7) significantly enhances performance, effectively enabling the SD model to address the matting task. Regarding the suggestion to switch to a weaker backbone, we note that the SDv1.5 model used in our experiments is already one of the weakest initialization versions comparing with current state-of-the-art SD models. Adopting other non-SD-based backbones would lack sufficient semantic information for the task.
> This clarification has been incorporated into **Sec. 4.5 of the revised manuscript**.
>
>
> We sincerely appreciate the insightful feedback from the reviewers and the Associate Editor, which has significantly improved the quality and clarity of our manuscript.
>
> Best regards,
>
> The Authors
>
>
>
> ### Reference
> [1] Hu, Y., et al.: Diffusion for natural image matting. In: ECCV, pp. 181–199 (2024)

---

> > ### Comment · Action_Editor_v4gu · 2025-10-11
> > **Reply**
> >
> > Thank you for providing the revised manuscript. The AE finds that the reviewers’ questions have been reasonably addressed. It would be beneficial to include a few more examples in Figure 8.
> >
> > AE